# Rare t(X;14)(q28;q32) translocation reveals link between *MTCP1* and chronic lymphocytic leukemia

Janek S. Walker [1,10], Zachary A. Hing[1,10], Steven Sher[1], James Cronin[1,2], Katie Williams[1], Bonnie Harrington[1], Jordan N. Skinner[1], Casey B. Cempre[1], Charles T. Gregory[1], Alexander Pan[1], Max Yano[1], Larry P. Beaver[1], Brandi R. Walker[1], Jadwiga M. Labanowska[3], Nyla A. Heerema[3], Krzysztof Mrózek[4], Jennifer A. Woyach [1], Amy S. Ruppert [1], Amy Lehman[5], Hatice Gulcin Ozer[6], Vincenzo Coppola [7,8], Pearlly Yan [1], John C. Byrd [1,9], James S. Blachly [1,6] & Rosa Lapalombella [1✉]

Rare, recurrent balanced translocations occur in a variety of cancers but are often not functionally interrogated. Balanced translocations with the immunoglobulin heavy chain locus (*IGH*; 14q32) in chronic lymphocytic leukemia (CLL) are infrequent but have led to the discovery of pathogenic genes including *CCND1*, *BCL2*, and *BCL3*. Following identification of a t(X;14)(q28;q32) translocation that placed the mature T cell proliferation 1 gene (*MTCP1*) adjacent to the immunoglobulin locus in a CLL patient, we hypothesized that this gene may have previously unrecognized importance. Indeed, here we report overexpression of human *MTCP1* restricted to the B cell compartment in mice produces a clonal CD5[+]/CD19[+] leukemia recapitulating the major characteristics of human CLL and demonstrates favorable response to therapeutic intervention with ibrutinib. We reinforce the importance of genetic interrogation of rare, recurrent balanced translocations to identify cancer driving genes via the story of *MTCP1* as a contributor to CLL pathogenesis.

---

[1] Division of Hematology, Department of Internal Medicine, The Ohio State University, Columbus, OH, USA. [2] Department of Veterinary Biosciences, The Ohio State University, Columbus, OH, USA. [3] Department of Pathology, The Ohio State University, Columbus, OH, USA. [4] The Ohio State University Comprehensive Cancer Center, Clara D. Bloomfield Center for Leukemia Outcomes Research, The Ohio State University, Columbus, OH, USA. [5] Center for Biostatistics, The Ohio State University, Columbus, OH, USA. [6] Department of Biomedical Informatics, The Ohio State University College of Medicine, Columbus, OH, USA. [7] Department of Cancer Biology and Genetics, The Ohio State University College of Medicine, Columbus, OH, USA. [8] Genetically Engineered Mouse Modeling Core, The Ohio State University and Arthur G. James Comprehensive Cancer Center, Columbus, OH, USA. [9] Division of Pharmaceutics and Pharmacology, College of Pharmacy, The Ohio State University, Columbus, OH, USA. [10] These authors contributed equally: Janek S. Walker, Zachary A. Hing. ✉email: rosa.lapalombella@osumc.edu

Chronic lymphocytic leukemia (CLL) is the most prevalent adult leukemia in Western countries and is characterized by a mature B-cell phenotype[1]. In contrast to other chronic adult leukemias, CLL is not primarily fusion/proto-oncogene driven. Instead, CLL pathogenesis likely begins with a lymphoid primed progenitor cell that becomes transformed at either a pre- or post-germinal center developmental stage bearing IGHV mutational status and a distinct epigenetic pattern corresponding to points along normal B-cell development[2–5].

During CLL transformation, loss or gain of genetic material then appears to be a key determinant of disease phenotype and clinical outcome, with chromosomal aberrations such as deletions in regions of chromosomes 11, 13, or 17 [del(11q23), (del(13q14), or del(17p13), respectively], or a gain in copy number of chromosome 12 (trisomy 12) observed in up to 80% of patients[6,7]. Alternatively, balanced translocations, specifically those including the immunoglobulin heavy chain locus (IGH; 14q32) resulting in constitutive overexpression of various proto-oncogenes in the B-cell compartment, occur far less frequently[8]. Despite their infrequence, molecular profiling of these rare rearrangements have revealed broad importance of previously un-recognized coding or non-coding genes critical to the pathogenesis of CLL. Practical application of this strategy facilitated understanding the role of the anti-apoptotic protein BCL2 in CLL. While abundantly present in follicular lymphoma and diffuse large B-cell lymphoma (DLBCL), the t(14;18)(q32;q21) translocation—involving the IGH locus and the BCL2 gene—is a rare event in CLL (1–2% of cases)[9,10]. Yet even in absence of a t(14;18)(q32;q21) rearrangement, it was found that BCL2 mRNA was over-expressed in virtually all CLL patients compared to normal B-cells[11]. Work to determine mechanisms driving this abnormality later revealed microRNAs miR-15/16 as leading posttranscriptional regulators of BCL2 and loss of miR-15/16 as a result of 13q14 deletions substantially associate with elevated BCL2 expression in CLL[12–14]. The pathogenic importance of this discovery subsequently led to a greater understanding of CLL disease biology and mechanisms resulting in the dramatic clinical activity demonstrated by the BH3-mimetic venetoclax[15]; ultimately contributing to the overall improvement in the therapeutic management of CLL.

Along this principle, we have identified a CLL patient with a previously undescribed t(X;14)(q28;q32) translocation, which leads to co-localization of the mature T cell proliferation 1 (MTCP1) coding region with the IGH locus, a situation analogous to the translocation of BCL2. Despite no known role in CLL, we found elevated MTCP1 mRNA expression in CLL cells compared to normal B cells and that increased MTCP1 expression in CLL patients portends poor outcomes to chemoimmunotherapy. Further interrogating this phenomenon we demonstrate the capacity for MTCP1 to initiate development of an aggressive murine CLL-like leukemia, revealing MTCP1 as a target for exploring the pathogenic mechanisms driving CLL. This discovery produces an in vivo model to investigate these unexplored mechanisms and to evaluate therapeutic strategies with optimal translatability to the clinical setting.

## Results

**t(X;14)(q28;q32) translocation identified in CLL.** The Xq28 locus containing MTCP1 is bicistronic, encoding two distinct transcripts as a result of an ancient insertion event[16]. MTCP1 (previously termed p13 MTCP1) lies within the first intron of an unrelated gene CMC4 (previously termed p8 MTCP1), with the two genes' distinct open reading frames preceded by a shared 5' UTR (Supplementary Fig. 1A). Previous studies, using X-ray diffraction to estimate the crystal structure of both p13 MTCP1 and p14 TCL1A proteins[17–19], describe a high degree of overlap between both amino acid sequence and 3-dimensional protein conformation—highlighted by an eight-strand beta barrel tertiary structure remarkably unique to this family of proteins (Supplementary Fig. 1B)[20–22]. Translocations involving the MTCP1 (Xq28) and T-cell receptor (TCRA/D) genes have been shown to induce constitutive MTCP1 overexpression and is a leukemic driving event in T-cell-prolymphocytic leukemia (T-PLL)[23–26]. To date, MTCP1 on the Xq28 locus has not been implicated in B-cell leukemia or lymphoma.

To explore a potential role for Xq28 translocations in B-cell malignancies, we screened metaphase karyotypes of 1744 cases suspected of CLL and identified eight (0.45%) with Xq28 rearrangements (Table 1). One case, a 59 year old female patient with IGHV-unmutated CLL, harbored a reciprocal t(X;14) (q28;q32) translocation, possibly involving the MTCP1 coding region and the IGH locus. To confirm this we performed fluorescent in-situ hybridization (FISH) analysis with probes directed against the IGH (3' red; 5'green) and MTCP1 (red) loci on metaphases. This showed that 5' IGH moved to the X chromosome (Fig. 1A) and MTCP1 was indeed translocated to the 5' end of the IGH locus (Fig. 1B). Combining the two probes demonstrated the MTCP1 and 3'IGH probes co-localized on a chromosome 14, consistent with the results with MTCP1 by itself, and 5' (green) IGH was on an X chromosome (Fig. 1C).

We then evaluated MTCP1 mRNA expression in CLL-B cells without any known Xq28 rearrangements and found ~2 fold higher MTCP1 mRNA transcripts in these CLL cells compared to naïve- or memory-B cells (Fig. 1D). A similar trend with CMC4 in CLL cells was also observed (Supplementary Fig. 1C). To interrogate the clinical significance of MTCP1 mRNA expression in CLL, we conducted a retrospective analysis on MTCP1 expression in CLL patients from two independent chemoimmunotherapy trial study cohorts for which microarray data have been previously reported (CALGB '9712' and '10101')[27,28]. Here, we correlated CLL risk factors with MTCP1 expression using baseline characteristics obtained at time of treatment initiation (Table 2). When stratified into quartiles by MTCP1 expression we observed a similar distribution between sexes and high-risk CLL cofactors including age, performance status, cytogenetic evaluation, IgHV mutation status, and Zap-70 methylation; with the exception of elevated blood lymphocyte counts (WBC) and advanced Rai stage at diagnosis skewing towards patients with higher MTCP1 expression (Q2-4). As a single continuous variable, a 2-fold increase in MTCP1 expression was also found to associate with shorter progression-free survival (PFS) when adjusting for study cohort (p = 0.03; Supplementary Table 1). To illustrate this association, we plotted PFS by MTCP1 expression quartile for all patients and for each study cohort; visualizing that patients with the lowest expression (Q1) tended to have longer PFS (Fig. 1E; Supplementary Fig. 1D, E). Further, as IgHV mutation status is a strong predictor of outcome in CLL[1], a bivariate analysis including continuous MTCP1 expression and IgHV status found a 2-fold increase in MTCP1 expression was prognostic for PFS independent of IgHV status (p = 0.03; Supplementary Table 2).

**B cell-specific MTCP1 overexpression drives a spontaneous and lethal leukemia.** Due to the proposed conservation of structure and function between p13 MTCP1 and p14 TCL1A, the fungible role of both in T-PLL pathogenesis, the known role of TCL1A in CLL[29], identification of Xq28 translocations in CLL with unresolved significance, and the apparent influence of MTCP1 expression on CLL outcomes, we hypothesized MTCP1 acts as a leukemogenic co-stimulator—revealing an unrecognized factor in CLL pathogenesis. To test this hypothesis, we generated

**Table 1 Clinical demographics for suspected CLL cases with Xq28 rearrangements.**

| Age (years) | Sex | Diagnosis | IGHV | Treated | Complexity | Karyotype[a] |
|---|---|---|---|---|---|---|
| 59[A] | F | CLL | Unmutated | No | 2 | 47,XX,+12[17]/47,idem,**t(X;14)(q28;q32)**[2]/46,XX[2].ish t(X;14)(5′ IGH+;3′ IGH+) |
| 59 | F | CLL | ND | Yes | 3 | 46,X,**t(X;12)(q28;q13)**,t(3;19)(q28;q13.1),add(20)(q11.2)[9]/46,XX[10]/nonclonal[2].ish t(3;19)(BCL6+, BCL3+;BCL6−,BCL3−) |
| 71 | M | CLL | Unmutated | yes | >6 | 46,XY,del(6)(q13)[cp2]/45,sl,der(6)ins(6;?)(p21;?)del(6)(q21),der(8)t(8;15)(p21;q15),-15,idic(16)(p11.2),add(17)(p11.2)[cp19]/45,**t(X;12)(q28;q13)**,-Y,+8,add(9)(q22),del(11)(q21),-22[cp2]/46,XY[2]/nonclonal[1].ish add(17)(TP53−) |
| 74 | F | CLL | Mutated | Yes | 4 | 46,XX,del(6)(q15q21)[9]/46,idem,**t(X;8)(q28;q22)**,inv(11)(p15q23)[cp3,one is 4n]/nonclonal w/clonal abnormalities[1]/46,XX[16] |
| 59 | M | Diffuse Large B-Cell Lymphoma | ND | No | >6 | 47,XY,der(X)**t(X;8)(q28;q24.2)ins(X;?)(q28;?)**,dup(1)(q21q42),dup(2)(q31q35),t(3;9)(q27;q32),del(7)(q22q32),der(8)**t(X;8)(q28;q24.2)**,+12,der(12)t(12;13)(q24.3;q22)x2,der(13)t(12;13)(q24.3;q12),t(14;19)(q32.3;q13.2)[cp16]/53,sl,+**der(X)t(X;8)ins(X;?)**,+4,+5,+del(7),+11,+der(12)t(12;13)[cp3]/46,XX[1] |
| 60 | M | CLL | Unmutated | Yes | >6 | 44,XY,del(9)(p22),psu dic(17;6)(p13;q21),dic(18;20)(p11.2;p11.2)[11]/45,sl,**add(X)(q28)**,+add(9)(p22),-del(9)(p22),dic(15;21)(p11.2;p11.2),+18,-dic(18;20),+20,+mar1[2]/90-91,sdl1x2,+**add(X)**,+**add(X)**,-Y,add(9)(q22)x4,-add(9)(p22),-add(9)(p22),+mar2[2]/44,sl,del(13)(q12q21.2)[5]/nonclonal w/clonal abnormalities[1] |
| 64 | F | CLL | Unmutated | yes | >6 | 60–79<4n>,XX,-X,-X,add(1)(q21)x2,add(3)(q29)x2,-4,-4,-6,-6,-8,-8,-9,-9,-13,-13,-14,psu dic(17;5)(p11.2;p13)x2,-18,add(18)(q23)x2,+mar1,+mar2x2[cp4]/72-80,sl,add(20)(q13.3),-mar1,+mar4[cp7]/58-78,sdl1,**add(X)(q28)**[cp2]/84-94,sdl1,+2,+7,+16,+19[cp2]/nonclonal w/ clonal abnormalities[5] |
| 56 | M | CLL | Unmutated | no | 4 | 45,Y,**der(X)(Xpter->Xq28::11q12->11q13::1q32->1qter)**,der(1)t(1;11)(q32;q13),**der(11)(?::11p15.5->11q12::Xq28->Xqter)**,-21[cp2]/46,XY[16]/nonclonal[4].ish der(X)(CCND1+),der(1)(ATM+) |

[a]Xq28 rearrangement indicated in bold.
[A]FISH analysis of CLL cells from this patient are depicted in Fig. 1.

a transgenic mouse model expressing recombinant human *MTCP1* under the control of a VH promoter-IgH-Eμ enhancer, targeting transgene expression to immature and mature B cells (Eμ-MTCP1; Supplementary Fig. 2A). Transgenic founders (Z36 and Z20) on the C57/BL6NTac background were bred to establish separate mouse lines, with successful passage of the *MTCP1* transgene to progeny confirmed via PCR (Supplementary Fig. 2B). To further confirm transgene integration into founder mice, we performed targeted locus amplification[30] in the established founder lines (Supplementary Fig. 2C, D). For both Z36 and Z20 founder lines, the genomic region between the 5′ and 3′ region of the integration site was deleted during the integration event. No structural variations were detected in the transgenic sequences, and >10 copies of the transgene integrated in each founder line.

CLL is characteristically defined as an accumulation of CD45+ B lymphocytes with phenotypic expression of cell-surface markers CD5, CD19, and CD23, with dim—or intermediate—expression of CD45R (B220)[1]. To evaluate the potential of the Eμ-MTCP1 model to develop a leukemia resembling a CLL-like phenotype, littermate mice from Z36 and Z20 founder lines were followed monthly via flow cytometry analysis of the blood. Gating on CD45+ cells and probing for CD19+/CD5+ and CD19+/B220dim cell populations, a progressive expansion of circulating CLL-like B-cells were detected as early as 5 months of age in both Eμ-MTCP1 founder lines (Fig. 2A). The percentage of transgenic mice spontaneously developing a CLL-like leukemia were 70% and 28% for founder lines Z36 and Z20, respectively (Fig. 2B). Using detection of >20% CLL-like cells in the blood as a threshold for leukemia onset, Eμ-MTCP1 founder lines Z36 and Z20

reached "diseased" status at a median time of 7.7 and 16.7 months, respectively (Fig. 2C). Regardless of type and severity of hematologic abnormality, the majority of Z36 and Z20 Eμ-MTCP1 mice died spontaneously or met early removal criteria (ERC) due to evidence of clinical deterioration at a median time of 10.8 and 14.5 months, respectively—a significant reduction from the median lifespan observed in wildtype littermates (Fig. 2D, E). Notably, the progressive accumulation of CLL-like B cells in Eμ-MTCP1 mice followed a similar trend to that observed in Eμ-TCL1 mice (Supplementary Fig. 3A)—a widely used CLL mouse model which reliably (reported 95–100% penetrance) develops a CLL-like phenotype driven by over-expression of recombinant human *TCL1A* also under control of a VH promoter-IgH-Eμ enhancer[31,32]. A cohort of Eμ-TCL1 mice in our laboratory maintained a similar median time to CLL-like leukemia onset with Eμ-MTCP1 founder line Z36 and a similar median survival with Eμ-MTCP1 founder line Z20 (Supplementary Fig. 3B, C). Evaluating only mice that developed a CLL-like phenotype by censoring animals at the time at which a T cell or myeloid cell abnormality was observed, a competing risk assessment estimated the median survival for founder lines Z36 and Z20 to be 12.4 months and 17.6 months, respectively (Supplementary Fig. 3D). While the estimated median survival of Eμ-MTCP1 founder Z20 extended beyond that observed in Eμ-TCL1 mice (14.1 months), we observed a significant reduction in time from leukemia onset to death in both Z36 and Z20 Eμ-MTCP1 founder lines (4.10 months and 2.94 months, respectively), when compared to the Eμ-TCL1 model (7.41 months; Supplementary Fig. 3E). Taken together, this evidence suggests that while the rate of CLL-like leukemia onset in Z20 lineage mice

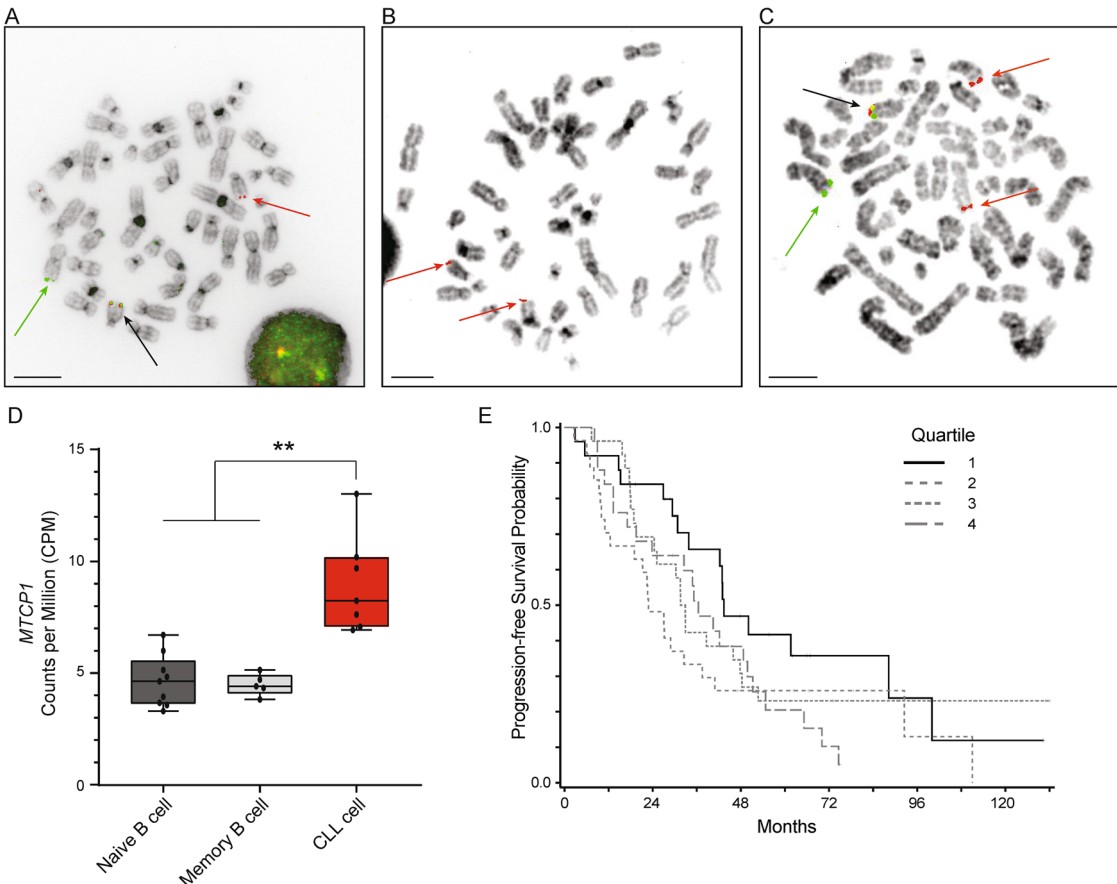

**Fig. 1 *MTCP1* is expressed in CLL patients and is associated with poor outcomes. A** Metaphase FISH was performed on CpG-stimulated CLL cells using the *IGH* break apart (3′red; 5′green) hybridization probes. Representative FISH analysis in one CLL patient harboring the t(X;14)(q28;q32) translocation showing the *IGH* probe has split with 5′ (green) on an X chromosome and 3′ (red) remaining on a chromosome 14. Images represent three independent tests. Scale bars are 10 μm. **B** Metaphase FISH was performed on CpG-stimulated CLL cells using the *MTCP1* (red) hybridization probe. Representative FISH analysis in the patient from (**A**) showing the *MTCP1* loci are on an X chromosome and on distal 14q. Images represent three independent tests. **C** Metaphase FISH was performed on CpG-stimulated CLL cells combining the *IGH* (3′red; 5′green) and *MTCP1* (red) probes. Representative FISH analysis in the patient from (**A**) showing the *MTCP1* and 3′*IGH* probes co-localize on a chromosome 14 and 5′(green) *IGH* is on an X chromosome. Images represent three independent tests. **D** *MTCP1* mRNA expression is elevated in CLL cells (without any known Xq28 rearrangements) compared to naïve- or memory-B-cell subsets ($p = 0.0012$, $p = 0.0013$, respectively). Data obtained from the Blueprint database. Box elements reflect 2nd - 3rd quartile, center line reflects the median value, and whiskers reflect the distance from the upper and lower limit to the box elements. CLL cells – red/right, $n = 7$; naïve B cells – dark gray/left, $n = 9$; memory B cells – light gray/middle, $n = 5$. P value estimated using a two-tailed unpaired $t$ test with Welch's correction. **E** Higher *MTCP1* expression in CLL patients identified with shorter progression-free survival (PFS) in chemoimmunotherapy trials. CLL patients from two independent study cohorts from Cancer and Leukemia Group B clinical trials (9712 and 10101; $N = 103$) were retrospectively analyzed. Patients were divided into quartiles: Q1 (solid line, $n = 25$), Q2 (medium-dash, $n = 27$), Q3 (short-dash, $n = 26$), and Q4 (long-dash, $n = 25$) according to *MTCP1* expression and PFS was visualized using the Kaplan–Meier method. As a continuous variable, the crude hazard ratio for a 2-fold increase in expression is 1.86 (95% CI: 0.97–3.55, $P = 0.06$) estimated from a Cox proportional hazards model. P value was determined via two-sided Wald test. No adjustments were made for multiple testing.

was significantly delayed in relation to Z36 lineage mice, upon CLL onset the kinetics of this CLL-like disease remained largely similar between the two founder lines.

A previously described hallmark of CLL B cells includes elevated expression of cytotoxic T-lymphocyte associated protein 4 (CTLA4)[33]. Our group has demonstrated that unless co-stimulation is provided, CTLA4 is not normally found on the surface of human CLL cells and expression is restricted to the intracellular compartment[34]. Using fluorescently labeled flow cytometry antibodies to detect both intracellular and surface CTLA4 in the blood from Eμ-MTCP1 mice, we found that, like human CLL and unlike Eμ-TCL1 CLL-like cells, CTLA4 expression was restricted to the intracellular compartment (Supplementary Fig. 3F).

**MTCP1 driven leukemia recapitulates aggressive human CLL.** Eμ-MTCP1 mice meeting ERC due to CLL-like disease invariably presented with splenomegaly accompanied by abdominal lympha-denopathy (Fig. 3A). Histopathology evaluation revealed variable neoplastic infiltration of lymphoid tissues including robust splenic and lymphatic involvement with very modest presence in the mar-row (Fig. 3B). Tumor cross sections exhibited variable degrees of F4/80 and B220-expressing infiltrates as detected by immunohis-tochemistry, with scant numbers of CD3⁺ mature, well-differentiated lymphocytes, which were interpreted as tumor-associated lymphocytes[35]. Healthy mouse tissues were infiltrated and effaced by neoplastic populations resembling both small-to-intermediate-sized lymphocytes and larger histiocytoid round cells, similar to those previously reported for the Eμ-TCL1 mouse model[31].

**Table 2 Association between *MTCP1* expression and baseline characteristics in CLL patients.**

|  | All Patients $N = 103$ | Quartile 1 Expression $n = 25$ | Quartile 2 Expression $n = 27$ | Quartile 3 Expression $n = 26$ | Quartile 4 Expression $n = 25$ | P value |
|---|---|---|---|---|---|---|
| Expression (Log 2) |  |  |  |  |  | NA |
| Median | 9.02 | 8.52 | 8.91 | 9.11 | 9.33 |  |
| Range | 8.16–10.26 | 8.16–8.70 | 8.78–9.02 | 9.03–9.19 | 9.20–10.26 |  |
| Study |  |  |  |  |  | 0.28 |
| 9712 | 41 (40%) | 7 (28%) | 9 (33%) | 12 (46%) | 13 (52%) |  |
| 10101 | 62 (60%) | 18 (72%) | 18 (67%) | 14 (54%) | 12 (48%) |  |
| Age |  |  |  |  |  | 0.07 |
| Median | 62 | 62 | 55 | 61 | 66 |  |
| Range | 34–83 | 42–81 | 34–77 | 38–83 | 46–79 |  |
| Hemoglobin |  |  |  |  |  | 0.08 |
| Median | 12.8 | 11.9 | 13.2 | 13.2 | 11.7 |  |
| Range | 5.5–16.9 | 5.5–15.1 | 9.9–16.9 | 8.2–15.5 | 6.6–15.1 |  |
| Missing/Unknown | 6 | 0 | 1 | 1 | 4 |  |
| WBC (count$^{E^3}$/μL) |  |  |  |  |  | 0.03 |
| Median | 108.6 | 67.0 | 132.0 | 108.6 | 131.0 |  |
| Range | 6.3–436.0 | 6.3–238.0 | 22.0–255.0 | 29.0–402.0 | 30.0–436.0 |  |
| Missing/Unknown | 3 | 0 | 1 | 0 | 2 |  |
| Sex |  |  |  |  |  | 0.66 |
| Male | 82 (80%) | 20 (80%) | 20 (74%) | 20 (77%) | 22 (88%) |  |
| Female | 21 (20%) | 5 (20%) | 7 (26%) | 6 (23%) | 3 (12%) |  |
| Performance Status |  |  |  |  |  | 0.29 |
| 0 | 61 (60%) | 14 (56%) | 12 (46%) | 18 (69%) | 17 (68%) |  |
| 1/2+ | 41 (40%) | 11 (44%) | 14 (54%) | 8 (31%) | 8 (32%) |  |
| Missing/Unknown | 1 | 0 | 1 | 0 | 0 |  |
| Rai Stage |  |  |  |  |  | 0.47 |
| I/II | 63 (61%) | 17 (68%) | 18 (67%) | 16 (62%) | 12 (48%) |  |
| III/IV | 40 (39%) | 8 (32%) | 9 (33%) | 10 (38%) | 13 (52%) |  |
| Cytogenetics Group |  |  |  |  |  | 0.18 |
| del(17p)/del(11q) | 22 (24%) | 7 (30%) | 7 (30%) | 2 (8%) | 6 (26%) |  |
| Other | 71 (76%) | 16 (70%) | 16 (70%) | 22 (92%) | 17 (74%) |  |
| Missing/Unknown | 10 | 2 | 4 | 2 | 2 |  |
| *IgHV* Usage |  |  |  |  |  | 0.77 |
| Mutated | 28 (31%) | 7 (30%) | 9 (38%) | 5 (23%) | 7 (33%) |  |
| Unmutated | 62 (69%) | 16 (70%) | 15 (63%) | 17 (77%) | 14 (67%) |  |
| Missing/Unknown | 13 | 2 | 3 | 4 | 4 |  |
| Zap-70 Methylation |  |  |  |  |  | 0.95 |
| <20% | 80 (79%) | 19 (76%) | 21 (78%) | 20 (80%) | 20 (83%) |  |
| ≥20% | 21 (21%) | 6 (24%) | 6 (22%) | 5 (20%) | 4 (17%) |  |
| Missing/Unknown | 2 | 0 | 0 | 1 | 1 |  |

Next, we analyzed the immunophenotypic signature of B-cell subsets using established murine markers of B-cell development. Representative plots are depicted in Fig. 3C, including comparison to blood from wildtype littermates and Eμ-TCL1 mice. After gating out CD3$^+$ T and CD11b$^+$ myeloid populations, cells were visualized using CD19 and CD5 expression markers: a phenotypically homogeneous CD19$^+$/CD5$^+$ co-expression population (CLL-like cells) with marked expansion in the blood was observed in Eμ-MTCP1 mice but not wildtype littermates. Typically, lymphocytes of significantly diseased Eμ-MTCP1 mice were composed of >60% CD19$^+$/B220$^{dim}$ B cells. CLL-like cells and maturing B cells (CD19$^+$/CD5$^-$) were further dissected according to CD21 and IgM expression to identify CD21$^+$/IgM$^+$ marginal zone/marginal zone progenitor (MZ/MZP) B cells, CD21$^{int}$/IgM$^{dim}$ follicular B cells, and CD21$^-$/IgM$^-$ or CD21$^-$/IgM$^+$ atypical B cells. Among the CLL-like cells, the frequencies of follicular and marginal zone/marginal zone progenitor cells were reduced in Eμ-MTCP1 mice compared to wildtype littermates. Instead, atypical B cells lacking CD21 expression were substantially increased in diseased Eμ-MTCP1 mice. These CD19$^+$/CD21$^-$ cells exhibited populations with varying degrees of IgM and IgD expression, pointing toward some heterogeneity within the bulk tumor population. Overall, the malignant cells of Eμ-MTCP1 mice showed a CD19$^+$/CD5$^+$/ CD93$^-$/B220$^{dim}$ phenotype with dim surface expression of IgM, IgD, and CD23, confirming a B1a cell phenotype. With respect to these surface markers, a similar trend between Eμ-TCL1 mice and wildtype littermates was observed.

**Eμ-MTCP1 leukemic cells are amenable to adoptive transfer in immunocompetent mice.** Given the spontaneous B-cell leukemia in the Eμ-MTCP1 model resembled human CLL, the ability to adoptively transfer these cells into immune competent murine hosts would further provide rationale for promotion of this model as a pre-clinical research tool. On this basis, we engrafted splenocytes isolated from an Eμ-MTCP1 mouse having reached euthanasia criteria due to progressive CLL-like expansion into 10 C57/BL6NTac mice (Fig. 4A). Successful engraftment into the immune competent hosts was evident by 3–10 weeks. Left to proliferate without pharmacologic intervention, engrafted mice succumbed to the progressive leukemic accumulation between 5 and 18 weeks post-adoptive transfer (Fig. 4B–D). Splenomegaly was always apparent, accompanied by abdominal lymphadenopathy. Histopathologic analysis showed development of a systemic lymphoid neoplasia comprising homogenous, small-to-intermediate lymphocytes affecting the spleen, lymph nodes, liver,

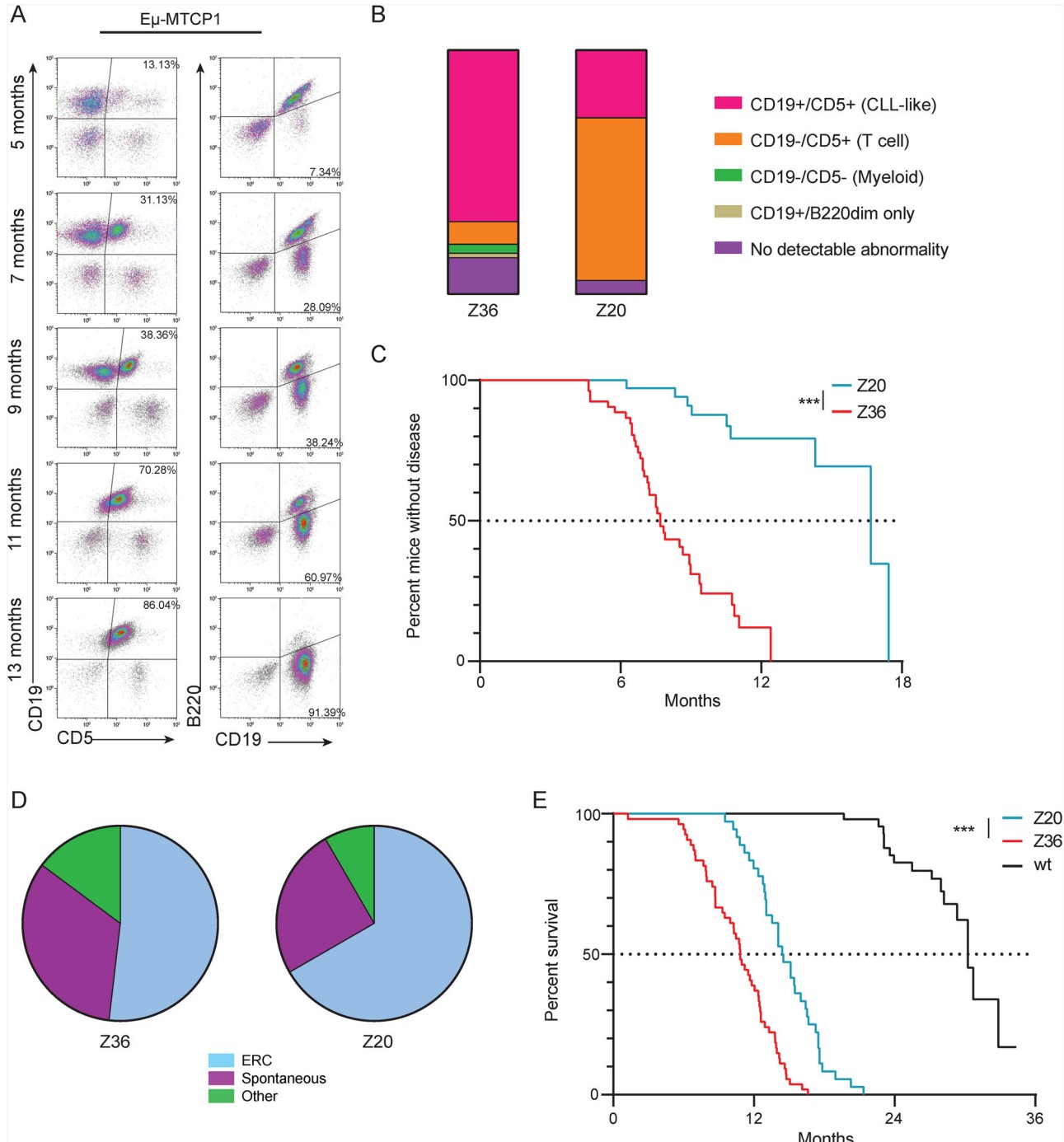

**Fig. 2 Overexpression of *MTCP1* drives a lethal CLL-like leukemia. A** Longitudinal flow cytometry analysis of a representative Eµ-MTCP1 mouse showing progressive development of a CD45+/CD5+/CD19+ and CD19+/B220dim CLL-like population in the blood. **B** Eµ-MTCP1 mice (Z36, n = 54; Z20, n = 36) were followed monthly by flow cytometry for disease progression as described in (**A**). Varying ratios of hemopathies were observed, including expansion of CD5+/CD19+ (CLL-like, pink) cells, CD5+/CD19− cells (T cells, orange), CD5−/CD19− (myeloid, green) cells, or CD19+/B220dim only CLL-like cells (tan). **C** Kaplan–Meier estimation of median time to disease onset in Eµ-MTCP1 mice (Z36 - red, n = 54; Z20 – blue, n = 36). Disease onset was defined as detection of >20% CD5+/CD19+ and CD19+/B220dim CD45+ cells in the blood determined by flow cytometry. Median time from birth to disease onset was shorter in Eµ-MTCP1 mice from the Z36 founder line (7.7 months) than from the Z20 line (16.7 months; p < 0.001). **D** Pie chart illustrating cause of death for Eµ-MTCP1 mice (Z36, n = 54; Z20, n = 36). Cause of death for a majority of Eµ-MTCP1 mice can be attributed to lethal progression of their disease burden. "ERC" = blue, mice met predefined early removal criteria. "Spontaneous" = purple, mice displayed disease progression identified by flow cytometry but had a spontaneous and unpredictable death. "Other" = green, mice died without measurable disease. **E** Kaplan–Meier estimation of median survival in Eµ-MTCP1 mice (Z36 - red, n = 54; Z20 - blue, n = 36). The median survival time was 10.8 months (95% CI: 9.5–12) for Eµ-MTCP1 founder line Z36 and 14.5 months (95% CI: 13.1–16) for founder line Z20 (p < 0.001). Representative survival of wildtype mice (black, n = 54) is shown as reference. P values (in **C** and **E**) determined by estimates from a Cox proportional hazards model.

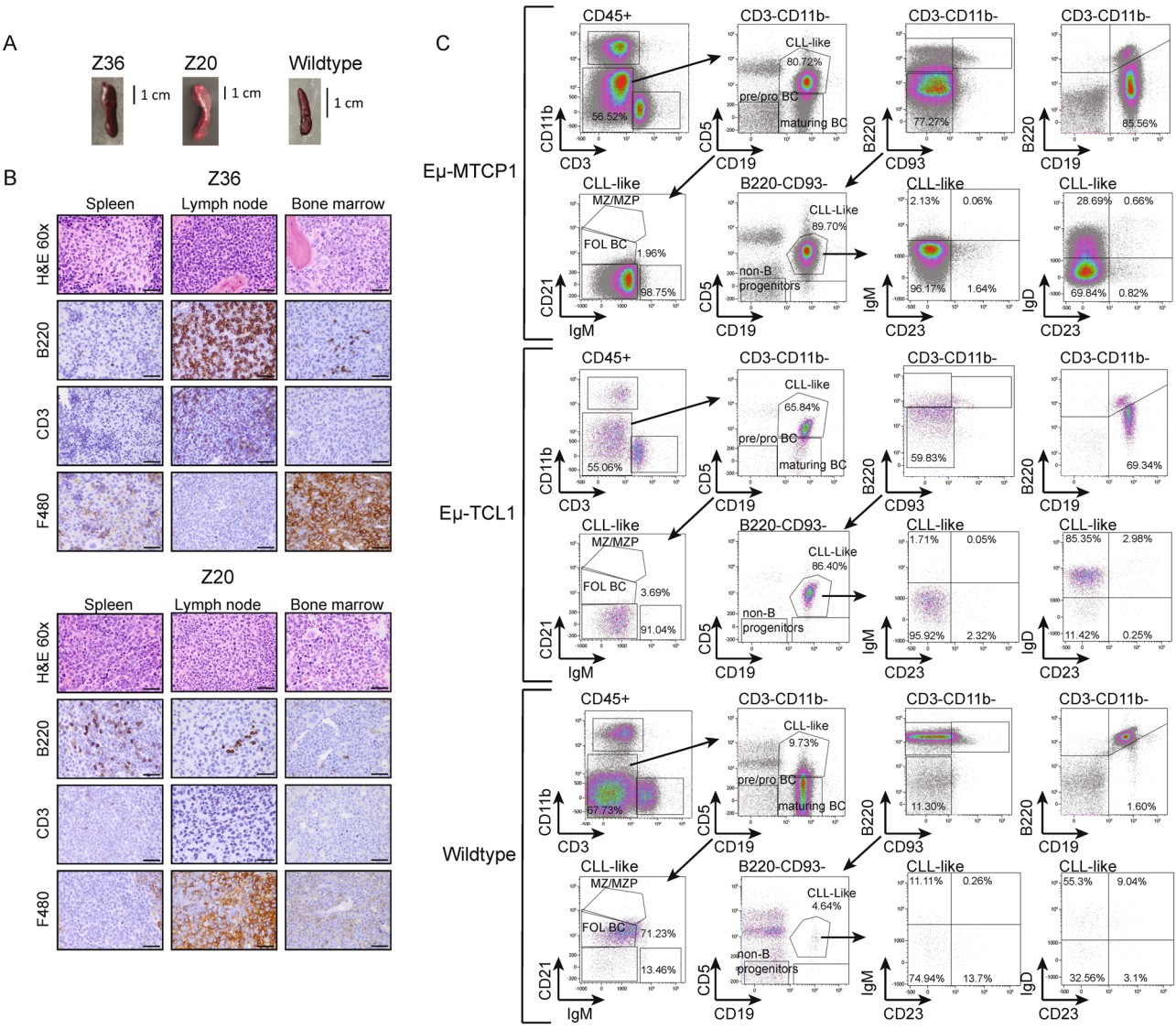

**Fig. 3 Accumulation of abnormal B lymphocytes in Eµ-MTCP1 mice. A** Representative gross images of severely enlarged spleens from Z36 and Z20 founder Eµ-MTCP1 mice, an observation consistently observed in subsequent progeny comprising the Eµ-MTCP1 colonies. A representative gross image of a healthy spleen from an age-matched wildtype mouse is shown for comparison. Scale bar is 1 cm. **B** Post-mortem histopathology analysis of cervical lymph nodes, bone marrow, and spleen from Eµ-MTCP1 founder mice Z36 and Z20 with CLL-like disease show histologic changes in multiple tissues consistent with a systemically disseminated mixed neoplasm containing B-lymphocytes and histiocytes. Lymphocytes were typically B220+, indicating a B-cell origin. CD3+ lymphocytes were sometimes scattered throughout the neoplasm and may be tumor infiltrating lymphocytes. Histiocytoid cells were identified as F4/80 positive in most cases. Cells null for B220, CD3, and F4/80 were occasionally observed. Z36 images representative of $n = 7$ evaluated mice, Z20 images representative of $n = 3$ evaluated mice. All organs visualized at 60x, scale bars are 33.3 µm. **C** Representative immunophenotypic evaluation of B-cell populations in blood derived from Eµ-MTCP1 (Z20 at ERC, 9 months old), Eµ-TCL1 (at ERC, 12 months old) and wildtype mice (12 month old). CD3+ T and CD11b+ myeloid cells were gated out and cells were plotted using CD19 and CD5 expression markers to identify CD19+CD5+ population (CLL-like cells) and CD19+CD5− (maturing B cells). CD19+CD5+ population were further dissected according to CD21 and IgM expression to identify CD21+IgM+ marginal zone/marginal zone progenitor (MZ/MZP) B cells, CD21intIgMdim follicular B cells, CD21−IgM− and CD21−IgM+ atypical B cells. CLL-like cells were also evaluated for IgD and CD23 expression. Overall, the malignant cells of Eµ-MTCP1 mice showed a CD19+CD5+CD93−B220dim phenotype with dim surface expression of IgM, IgD, and CD23 confirming a B1a cell phenotype.

bone marrow, Peyer's patches, thymus, serosal surfaces of viscera; and rarely, lung and kidney (Fig. 4E). Thus, while a mixed lineage neoplasia was observed in Eµ-MTCP1 founder lines with a CLL-like leukemia, only the B-lymphoid portions of the total leukemic burden successfully engrafted and continued to proliferate in the host.

**CLL-like transcriptional profile identified in CD19+CD5+ populations from Eµ-MTCP1 mice.** To understand if the

aggressive murine leukemia in Eµ-MTCP1 transgenic mice were composed of heterogeneous, polyclonal B cells or derived from a single precursor founding a homogeneous tumor population as is the case in human CLL, we evaluated the B-cell receptor (BCR) repertoire by examining *IGH* transcripts via RNA-sequencing methods previously described by our group[36]. Prominent usage of distinct heavy chain gene loci representing a clonal B cell expansion was observed from splenic B cells isolated from 3/3 Eµ-MTCP1 mice and similarly in 3/3 Eµ-TCL1 mice, a stark contrast from the high degree of clonal variability exhibited in wildtype

mice (Fig. 5A, Supplementary Table 3). The major clone in 2/3 Eμ-MTCP1 and 3/3 Eμ-TCL1 mice predominantly used $V_H1$ and $V_H12$ family genes, while 3/3 wildtype mice predominantly used $V_H5$ family genes. One analyzed Eμ-MTCP1 mouse used $V_H5$ family genes in its dominant clones. The $V_H12$-3 gene was shared as that predominantly used in the major clone of one Eμ-MTCP1 and Eμ-TCL1 mouse. The most abundant tumorigenic clones isolated from Eμ-MTCP1 spleens exhibited a low *IGHV* mutational burden (Supplementary Fig. 4A), well below the threshold for classification[37] as "mutated" (Supplementary Fig. 4B). The low mutational burden in this region is consistent with the aggressive, *IGHV*-unmutated, subtype of human CLL.

We next sought to determine the overall transcriptional profile of these monoclonal tumor cells. Principal component analysis of global transcription profiles revealed significant segregation of splenic B-cells collected from Eμ-MTCP1 and Eμ-TCL1 mice, and splenic B-cells from wildtype littermates across PC1 (Fig. 5B). Further, a significant overlap in gene expression most variable from wildtype splenic B-cells was found when evaluating Eμ-MTCP1 or Eμ-TCL1 transgenic strains, suggesting the leukemic B1a cells in Eμ-MTCP1 mice have achieved a transformed state by both phenotypic and transcriptomic standards (Fig. 5C, D). Ingenuity pathway analysis (IPA) of significantly enriched genes (Log2FC > 2, $p < 0.001$) shared between Eμ-MTCP1 and Eμ-TCL1 transgenic strains when compared to wildtype littermates primarily converged on genes involved in G protein-related signaling pathways (Supplementary Fig. 5A), likely a consequence from both MTCP1 and TCL1 acting as activators of AKT. In genes uniquely enriched in Eμ-MTCP1 mice compared wildtype littermates, IPA analysis converged on FAK and PTEN signaling, Rho family GTPase signaling, and protein kinase A/cAMP-mediated signaling (Fig. 5E). Directly comparing Eμ-MTCP1 and Eμ-TCL1 mouse transcriptomes revealed a considerable degree of similarity, where only 79 of 15,318 analyzed genes displayed significant variation from one transgenic model to the other (Fig. 5F, Supplementary Fig. 5B). Specifically, 38 genes were identified as overrepresented in Eμ-TCL1 mice, highlighted by the presence of various nucleotide binding factors (*Atrip*, *Ddx60*, *Trim6*), heat shock proteins (*Hspa1a/b*), and other known oncogenic markers (*Adm*, *Eef2k*, *Il12a*, *Ly6i*, *Mapk8*, *Map3k20*, *Pmaip1*, *Wnt16*). Relative overexpression of 41 genes was noted in Eμ-MTCP1 mice, marked by the presence of cell signaling molecules (*Ccr10*, *Lrp5*, *Zbtb4*) and other known oncogenic markers (*Cd34*, *Ptgs1*). Notably, no change in *Tcl1* transcript abundance was observed between Eμ-MTCP1 and wildtype mice, and no change in *Mtcp1* transcript abundance was observed between Eμ-TCL1 and wildtype mice (Supplementary Fig. 5C), suggesting the *TCL1*-driven murine leukemia acts independently from *MTCP1* and the *MTCP1*-driven murine leukemia acts independently from *TCL1*.

**Ibrutinib treatment impairs Eμ-MTCP1 leukemia development.** We previously demonstrated the significance of constitutive activation of BCR signaling pathways in promoting proliferation of CLL cells, highlighting efficacy of the Bruton tyrosine kinase (BTK) inhibitor ibrutinib in abrogating leukemic advancement even in pre-leukemic stages[38]. To evaluate the sensitivity of the MTCP1-driven murine leukemia to ibrutinib, Eμ-MTCP1 pups from founder line Z36 were treated continuously from the time of weaning with drinking water containing ibrutinib at ~30 mg/kg/day or 10% cyclodextrin (vehicle). Monthly assessment of circulating $CD5^+/CD19^+$ and $CD19^+/$ B220$^{dim}$ leukemia cells in all treated mice displayed a lower leukemic burden at 6 and 12 months, contributing to considerable survival prolongation in mice receiving ibrutinib at

12 months of age (Fig. 6A, B). We further evaluated the use of the Eμ-MTCP1 adoptive transfer model for pre-clinical evaluation of CLL drug candidates. Successfully engrafted mice with comparable disease load (percent $CD19^+/CD5^+$ B cells in the blood) were randomly assigned to receive ibrutinib or vehicle by daily oral gavage at any given enrollment time to control for different growth kinetics of the engrafted tumor cells. Similarly, ibrutinib administration led to a reduction in the rate of disease development between six and 12 weeks post-enrollment and prolonged survival compared to those receiving vehicle (Fig. 6C, D). Postmortem histopathology analysis supported these evaluations, where ibrutinib-treated mice displayed less-severe organ involvement (Fig. 6E).

## Discussion

Here, we identified an index CLL patient bearing an uncommon, reciprocal t(X;14)(q28;q32) translocation joining the *MTCP1* locus with immunoglobulin heavy chain regulatory elements (*IGH*; 14q32), an event analogous to the *IGH* translocations with *MYC*, *CCND1*, or *BCL2* genes driving various B-lymphomas[39,40]. After observing that this translocation occurs in CLL, we further discovered that even without an Xq28 rearrangement *MTCP1* mRNA is overexpressed in CLL cells as compared to normal B-cells. The observed overexpression of *MTCP1* in CLL and the discovery of a translocation that juxtaposes with the *IGH* locus suggest a pathogenically relevant role of *MTCP1* in CLL. Notably, increased *MTCP1* expression in this CLL cohort lacked major correlation with pre-treatment characteristics but was associated with a shorter response to chemoimmunotherapy. This relationship may reflect the current understanding of *TCL1* expression in CLL, where inter-patient variability remains the norm yet patients with the lowest *TCL1* expression maintain favorable outcomes following chemoimmunotherapy[41,42]. Further correlating *MTCP1* expression with PFS, a bivariate analysis identified a 2-fold increase in *MTCP1* expression as a prognostic indicator of PFS independent of *IgHV* status, a known high-risk factor in CLL. Importantly, however, *IgHV* mutation status as a single variable was not prognostic for reduced PFS in this chemoimmunotherapy cohort—which is atypical in CLL—suggesting further large scale correlative studies are necessary to provide a comprehensive assessment of the associations between higher *MTCP1* expression and other high-risk factors in CLL.

In a multivariate analysis, a 2-fold increase in *MTCP1* expression lost strong association with reduced PFS while adjusting for additional predictors for reduced PFS in these chemoimmunotherapy trials (Zap-70 methylation, high-risk cytogenetics, sex, WBC). This result implicates *MTCP1* as a factor with considerable influence on the CLL disease course; yet it may not act as the fundamental driving element, which is not entirely surprising considering the abundance of evidence conferring the significant relationship between CLL outcomes and these other high-risk factors. Even so, revealing mechanisms supporting *MTCP1* upregulation in CLL may provide meaningful insight to the overall pathogenesis of this disease. Similar to studies evaluating *TCL1* in CLL[43–45], the apparent inter-patient variability suggests multiple factors likely contribute to dysregulation of *MTCP1* expression; where loss of microRNA-mediated negative regulation or *MTCP1* upregulation via stimulating signals from the tumor microenvironment are primary candidates for further evaluation. Likewise, exploring BCR-induced kinase cascades, with particular attention given toward the role for PKC-β[46], may facilitate understanding of the upstream mechanisms upregulating and activating *MTCP1* in CLL. Other mechanisms supporting *MTCP1* upregulation, such as loss of epigenetic control or evasion of X chromosome inactivation (although we

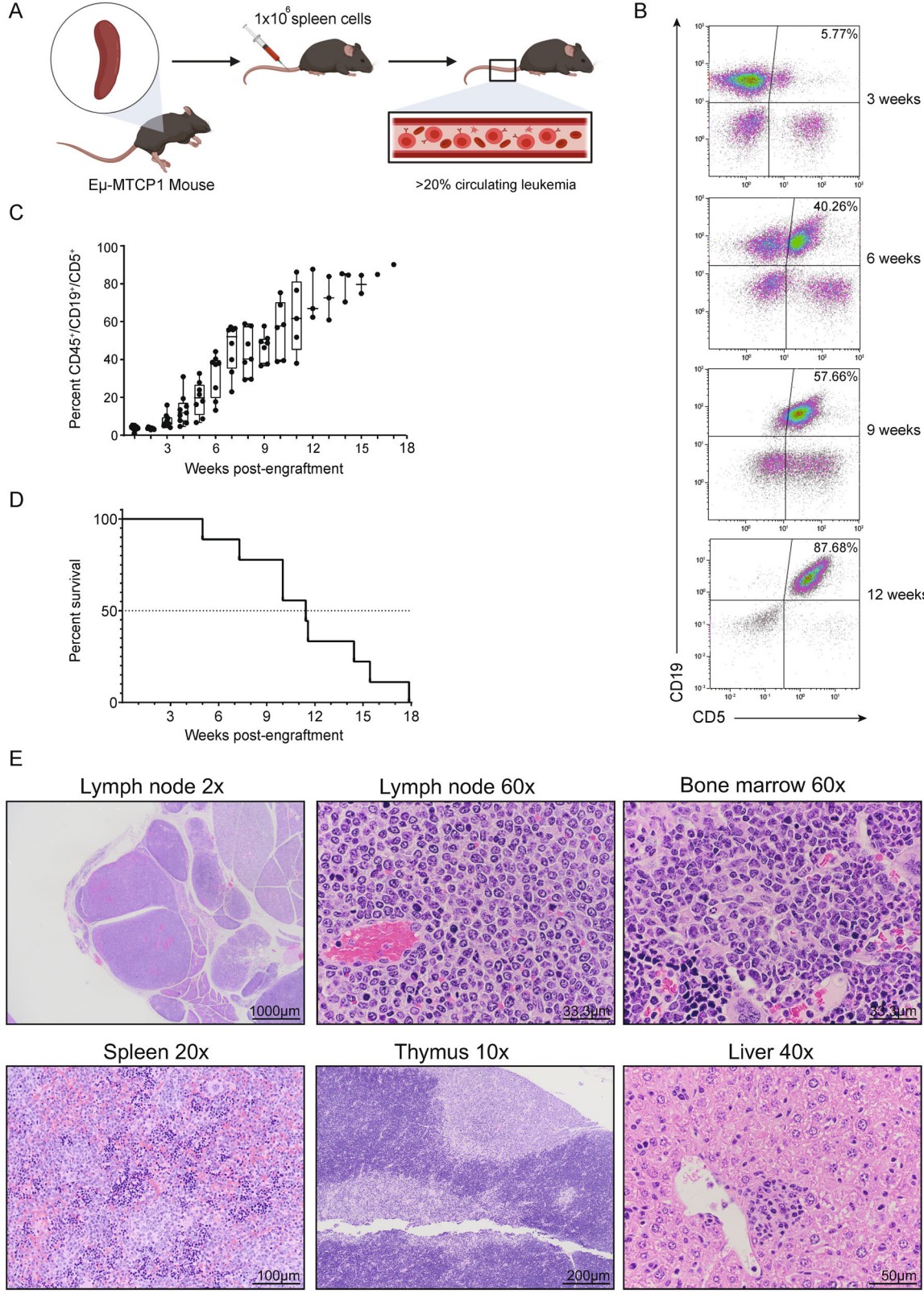

observed an even male:female distribution between *MTCP1* expression quartiles) serve as additional unexplored means facilitating *MTCP1* upregulation in CLL. In addition, exploration of this gene in other diseases such as acute myeloid leukemia (AML) might be considered based upon the recent identification of a t(X;17)(q28;q21) rearrangement resulting in a *KANSL1-MTCP1* fusion gene in an AML patient[47].

Supporting the notion that *MTCP1* expression beyond basal levels is advantageous for leukemogenic B cells, screening a large cohort of suspected CLL cases revealed seven additional Xq28 rearrangements with unexplored relevance; two with a translocation at the 12q32 site, one each involving 8q22 or 8q24.2 sites, and three joining unrecognized material with the Xq28 site. Microdeletions in the 12q32 site including the *HOXC* cluster and

**Fig. 4 The CLL-like disease populating the spleen of Eµ-MTCP1 mice is amenable to adoptive transfer. A** Schematic review of Eµ-MTCP1 adoptive transfer leukemia model. $1 \times 10^6$ splenocytes isolated from an Eµ-MTCP1 mouse (from founder line Z36) having reached predefined euthanasia criteria due to progressive leukemic expansion were engrafted via tail vein injection into immunocompetent C57/BL6NTac (wildtype) mice. **B** Immunocompetent recipient mice described in (**A**) were monitored weekly for expansion of CD5+/CD19+ and CD19+/B220dim circulating CD45+ cells by flow cytometry. A representative flow analysis is shown describing the progressive accumulation of CD45+/CD5+/CD19+ cells. **C** Nine of ten recipient mice displayed significant peripheral disease (CD45+/CD5+/CD19+ cells) between 3 and 10 weeks post-adoptive transfer of splenic cells from Eµ-MTCP1 mice. In mice surviving at least 12 weeks post-engraftment, the engrafted tumor population nearly composes the entire peripheral blood compartment (>75%). Box elements reflect 2nd - 3rd quartile, center line reflects the median value, and whiskers reflect the distance from the upper and lower limit to the box elements. **D** Kaplan–Meier estimation of median survival in recipient adoptive transfer mice. Without intervention, adoptive transfer of splenic CLL-like cells from Eµ-MTCP1 mice results in a lethal disease with a median time to disease at 11.4 weeks. All successfully engrafted mice eventually succumbed to disease. **E** Representative post-mortem histopathology analysis of lymph node (2x, 60x), bone marrow (60x), spleen (20x), thymus (10x), and liver (40x) from a mouse in the Eµ-MTCP1 adoptive transfer model reaching ERC due to progressive disease. Hematoxylin & eosin staining shows histologic changes consistent with a systemically disseminated round cell neoplasm of small to intermediate lymphocytes that was homogenous between engrafted animals. Images representative of $n = 3$ mice. Scale bars as indicated.

other adjacent genes results in significant disruption of normal cell function[48–50], and amplification of the 8q22 site including *EDD1* and *GRHL2* genes, negative regulators of apoptosis, have been described in breast, pancreas, and lung cancer cells as a mechanism for evasion of death receptor-activated therapies[51]. The 8q24.2 region is a gene desert containing *MYC* enhancer elements and variations at this site are known to influence CLL[52]. Interestingly, the t(X;8)(q28;q24.2) case reported here was identified in a patient screened initially on the basis of suspected CLL which was later found to be DLBCL. Cytogenetic evaluation of this patient, a 59 y/o male, also identified a t(3;9)(q27;q32) rearrangement involving the *BCL6* gene and a t(14;19)(q32.3;q13.2) rearrangement involving the *BCL3* gene. This t(14;19) rearrangement, joining the *BCL3* locus with *IGH* elements, occurs in DLBCL and other chronic lymphoproliferative disorders but is most frequently observed in CLL. However, t(14;19) CLL cases are often atypical with distinctive clinicopathologic and genetic features including younger age, aggressive clinical course, and association with trisomy 12, which was also present in this patient[53,54]. Translocation of the *MTCP1* locus under *MYC* enhancer elements in this unusual case supports our finding that, while rare, genomic rearrangement events resulting in up-regulation of the *MTCP1* gene may contribute to the transformative potential of the leukemic B cell and influence the overall trajectory of the resulting tumor burden. Juxtaposition of the *MTCP1* gene locus with these additional sites is of considerable interest and further investigation will be required to fully describe their significance.

To establish a definitive role for *MTCP1* as a pathogenic contributor in CLL, we generated a mouse model with B cell-specific overexpression of human recombinant *MTCP1* (Eµ-MTCP1). Longitudinal evaluation of Eµ-MTCP1 littermates revealed a majority of these mice developed a lethal hematologic malignancy, highlighted by the progressive emergence of CLL-like B cells or hyperproliferative T cells circulating in the blood and accumulating in the spleen and lymph nodes. The CLL-like population in Eµ-MTCP1 mice bears a striking resemblance to the disease that develops in Eµ-TCL1 mice; however, the timeline for clinical deterioration from CLL-onset to death was accelerated in both Eµ-MTCP1 founder strains. The observed disparity in median estimated survival between Eµ-MTCP1 founder strains Z20 and Z36 appears to be largely driven by the delayed rate of leukemia onset in Z20 lineage mice. While we confirmed >10 copies of the *MTCP1* transgene were inserted in both founder lines, we acknowledge the integration site occurred at two distinct locations on different chromosomes for each respective founder. While mechanisms supporting the observed variance in leukemia onset between Z20 and Z36 founder lines remain unresolved,

insertion of the *MTCP1* transgene in proximity to additional active enhancer regions or inaccessible in regions of condensed chromatin may be contributing to this distinction. Shared between founder lines, however, the cause of death for a majority of Eµ-MTCP1 mice was attributed to disruption of critical organ function due to systemic malignant infiltration comprised primarily of mixed populations of small-to-intermediate-sized lymphocytes and larger histiocytoid cells. Similar to what is often observed in CLL patients, the spleens of Eµ-MTCP1 mice were deformed, consistently presenting with substantial splenomegaly.

Further analysis of B lymphocytes from Eµ-MTCP1 mice revealed that this CLL-like disease may present a faithful resemblance to human CLL. We found that circulating CLL cells from Eµ-MTCP1 mice had intracellular, but lacked surface expression of the immunomodulatory molecule CTLA4, consistent with human CLL cells but not Eµ-TCL1 CLL-like cells[34]. Immunophenotypic analysis of the CD19+/CD5+ cells of Eµ-MTCP1 mice revealed the vast majority were B220low/CD93− expressing cells, suggesting a B1a cell phenotype. Scant numbers of IgM+ or IgD+ were found within this population, corroborating histopathology reports suggesting some degree of heterogeneity is evident within the malignant population. Engraftment of bulk splenocytes from Eµ-MTCP1 mice with CLL-like disease into immune competent hosts resulted in a homogeneous expansion of tumorigenic B lymphocytes, suggesting the true tumor population is comprised of the clonally related B1a cells and the chronic inflammatory milieu produced by the proliferating B-cells in Eµ-MTCP1 mice may promote proliferation of macrophages, plasma cells, and extramedullary hematopoiesis.

Having demonstrated that the Eµ-MTCP1 mouse model generates a CD19+/CD5+ B-cell malignancy with resemblance to human CLL, we proposed that this model may be ideally suited for pre-clinical evaluation of therapeutic agents for consideration in CLL and related diseases. To support this proposal, we demonstrated that continuous ibrutinib administration in pre-leukemic mice delayed disease onset and prolonged survival. In an adoptive transfer model, daily ibrutinib dosing delayed the rapid advancement of engrafted Eµ-MTCP1 splenocytes and resulted in a dramatic prolongation in survival. The moderate reduction in circulating lymphocytes upon treatment with ibrutinib in this adoptive transfer model is consistent with the understanding that inhibition of BCR signaling via ibrutinib drives cell mobilization from nodal sites and often results in prolonged lymphocytosis[55]. The dramatic prolongation in survival seen here is likely a result of less severe occupation of proliferative lymphoid compartments and defacement of normal organ architecture in engrafted mice. Overall, the observed

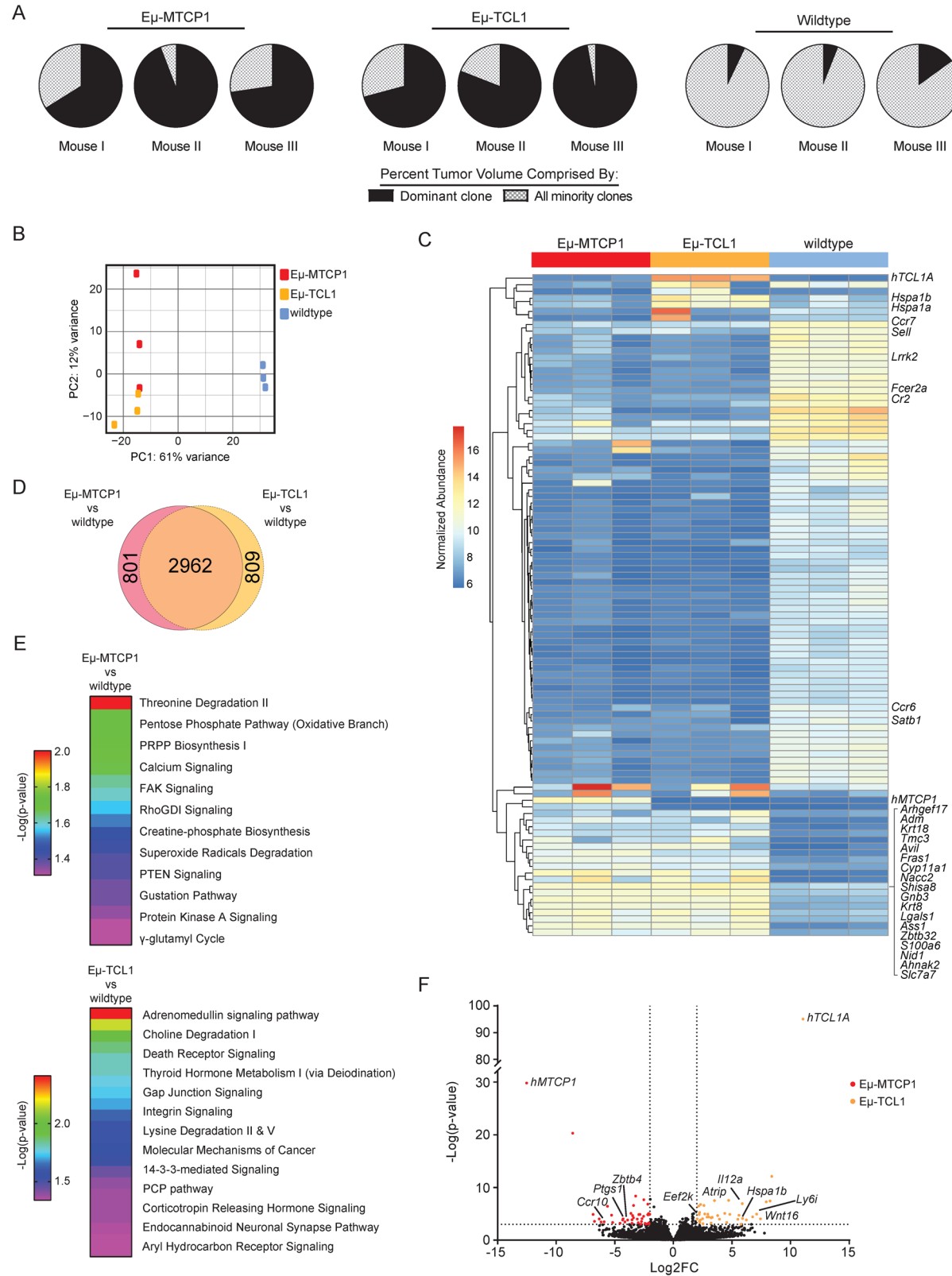

response to inhibition of BCR signaling encourages further use of this model in development of novel derivatives with varying specificity for BTK and other components of the BCR pathway.

Although the overall transcriptome profile remained largely similar between Eμ-MTCP1 and Eμ-TCL1 tumor cells, further investigation to define the significance in variation between *MTCP1-* and *TCL1*-driven CLL is warranted. Specifically,

elucidating the leukemogenic mechanisms which ultimately led to an accelerated disease course from CLL-onset to death in Eμ-MTCP1 mice is of considerable interest but is beyond the scope of the present study. Furthermore, the conserved structure between MTCP1 and TCL1A proteins and similarities in the resulting murine and human CLL phenotype suggest shared activation of AKT and other leukemogenic pathways; pathways which may be

**Fig. 5 Transcriptome profiling reveals transformed B cells in Eμ-MTCP1 mice. A** Comparison of *IGHV* gene usage among Eμ-MTCP1 (founder Z36), Eμ-TCL1, and wildtype mice (n = 3 per group) demonstrated clonal tumor populations in Eμ-MTCP1 and Eμ-TCL1 mice. Visualized via pie chart, the percent tumor volume comprised by the dominant clone of each mouse shown in black. All minority clones are grouped and their percent contribution is visualized via gray-checkered pattern. Gene names of the top 10 *IGH* genes in order of abundance are presented in Supplementary Table S3. **B** Unbiased RNA-sequencing of splenic B-cells isolated from Eμ-MTCP1 (founder Z36), Eμ-TCL1, and wildtype mice (n = 3 per group) revealed distinct clustering of Eμ-MTCP1 (red) and Eμ-TCL1 (yellow) samples from wildtype mice (blue) as visualized by principle component analysis. 62% variance was observed along PC1, and 12% variance was observed along PC2. **C** The 100 most-variable genes from (**B**) visualized via heatmap. A majority of these significantly differing genes encode *IGH* genes (not shown for clarity—whole list of genes can be found in the source data). Genes listed on the y axis are arranged via hierarchical clustering using Euclidean distance measurements. Heat map scale is representative of normalized Log2 expression between all samples. **D** Euler diagram visualizing differentially expressed genes between Eμ-MTCP1 vs wildtype mice (left circle, pink) and Eμ-TCL1 vs wildtype mice (right circle, yellow). A significant overlap (2962/4572; 65%) in the number of differentially expressed genes compared to wildtype mice were shared between Eμ-MTCP1 and Eμ-TCL1 mice (middle overlap, orange). **E** Ingenuity pathway analysis (IPA) of significantly enriched genes (Log2FC > 2, p < 0.001) unique to Eμ-MTCP1 vs wildtype mice or Eμ-TCL1 vs wildtype mice. Rainbow color scale reflects –Log(p value) determined via Fisher's exact test. Complete list of IPA terms and p values are available in the Supplementary File: source data. **F** Differential mRNA expression analysis of splenic B cells from Eμ-MTCP1 and Eμ-TCL1 mice visualized via volcano plot (n = 3 per group). Seventy-nine genes enriched in Eμ-MTCP1 mice (red) or Eμ-TCL1 mice (yellow) are shown (Log2FC > 2; p < 0.001). P value determined in DESeq2 via Wald test. Complete list of genes and p values are available in the Supplementary File: source data.

independently activated as no observable change in *Tcl1* expression was found in Eμ-MTCP1 mice and no observable change *Mtcp1* expression was found in Eμ-TCL1 mice. Validation of this relationship remains unreported, however, for which the Eμ-MTCP1 mouse presents an excellent tool to study these mechanisms. The transgenic strategy of the Eμ-MTCP1 mouse presented herein also supports the notion that the *CMC4* gene is merely a passenger in the t(X;14)(q28;q32) rearrangement and may not provide an essential role in the leukemogenic transforming event. Regardless, *CMC4* expression was indeed elevated in CLL cells when compared to normal B-cell subsets. When translated to the short p8 MTCP1 isoform this protein is localized to the mitochondria[56], likely suggesting an indirect role in supporting leukemogenesis via metabolic pathways. Thus, complete investigation of the relationship between p8 and p13 MTCP1 may further define the collective oncogenic impact of Xq28 rearrangements.

A broader and more general finding relevant to other types of cancer is the successful application of a strategy pursuing the functional consequence of genes involved in rare chromosomal abnormalities. Exploring the rare t(14;18)(q32;q21) translocation in CLL, such approaches facilitated understanding of the significance of the anti-apoptotic protein BCL2 and *miR-15/16* at the minimal deleted region of del(13q14)[12–14]. These microRNAs contribute to the pathogenesis of CLL, and BCL2 represents an exceptionally valuable therapeutic target[15]. Whole genome sequencing may effectively identify translocations involving *BCL2* and *IGH*[57], but the t(X;14)(q28;q32) rearrangement has not been previously reported. This and other Xq28 translocations may have escaped notice for several reasons. Identification of non-fusion gene translocations [i.e., precisely the situation in t(v;14)(v;q32) translocations] may be hampered by fundamental limitations of short-read sequencing, namely poor mappability of highly repetitive regions flanking translocation breakpoints. In addition, the infrequent occurrence of these Xq28 changes in the very large cytogenetics cohort in our series—as compared to the largest WGS study of CLL[57] is another potential explanation. Regardless, our experience suggests an opportunity for both classic cytogenetics and WGS assessment of rare recurrent balanced translocations to provide an avenue in other cancers to identify previously unrecognized oncogenes.

In summary, evidence from both human CLL and a transgenic mouse model present a causal relationship between *MTCP1* and CLL. The Eμ-MTCP1 mouse model should be considered as an alternative tool for both biologic assessment of co-expressed

genes and pre-clinical evaluation of CLL therapeutics. Identification of *MTCP1* as a significant factor in the pathogenesis of CLL provides an additional molecular target for consideration in this disease, and an archetypical process for the future pursuit of rare balanced translocations for the identification of genes relevant to pathogenesis and progression in many cancers.

## Methods

**Screening suspected CLL cases for Xq28 rearrangements**. A total of 1744 suspected CLL specimens collected between November 2003 and December 2014 at The Ohio State University Comprehensive Cancer Center were evaluated for possible Xq28 rearrangements by screening metaphase karyotypes collected at time of initial biopsy. Translocations involving Xq28 were confirmed by two independent cytogeneticists.

*MTCP1* and *CMC4* gene sequences were obtained and visualized from the Ensembl genome browser. The proposed 3-dimensional crystal structure for MTCP1 and TCL1A protein sequences were obtained from the RCSB Protein Data Bank (IDs: 1A1X, 1JSG, respectively)[18,19]. Crystal structures were determined via X-ray diffraction and presented at 2.00 Å and 2.50 Å resolution, respectively. Gene expression data for *MTCP1* and *CMC4* determined from RNA-sequencing were collected from the Blueprint DCC data portal[58]. For evaluation of *MTCP1* expression via RNA-sequencing, de-identified human CLL cells were isolated as previously described after obtaining informed consent on protocols approved by The Ohio State University Cancer Institutional Review Board.

**Fluorescent in-situ Hybridization**. Fluorescent in-situ Hybridization (FISH) was performed with IGH/CCND1 XT, IGH break apart (Abbott Molecular, Downers Grove, IL) and MTCP1 (Empire Genomics, Williamsville, NY) probes. FISH was done according to the manufacturer's recommendations, except prior to hybridization slides were pretreated with pepsin and postfix solution. Co-denaturation of probe and sample was done on HyBrite (Abbott Molecular, Downers Grove, IL) for 5 min at 73 C. Hybridization was carried out overnight at 37 C, and slides were washed in 0.4 x SSC/0.3%NP-40 for 2 min at 73 C. The signals were viewed using a fluorescent microscope (Zeiss Axioscope 40) equipped with appropriate filters and analyzed with Applied Imaging System.

**Generation of the Eμ-MTCP1 mouse model**. Transgenic Eμ-MTCP1 mice were generated on a C57BL/6NTac background at The Ohio State University Comprehensive Cancer Center's Transgenic Mouse Facility via pronuclear injection of linear constructs derived from a plasmid vector encoding murine immunoglobulin mu enhancer elements followed by human cDNA encoding the p13 kDa MTCP1 protein. An equal ratio of male and female mice were maintained throughout all analyses.

From five unique transgenic founder lines, detailed characterization was conducted in two lines showing early evidence of a disease phenotype (Z36 & Z20). Genotyping of Eμ-MTCP1 progeny was performed using the following *MTCP1* primer sequences: (forward: 5′ ATCTGCCGCCACCATGGC 3′; reverse: 5′ GCT TAAGCAACAGCTCCTGTAC 3′). All experiments were carried out under protocols approved by The Ohio State University Institutional Animal Care and Use Committee. Pre-defined euthanasia criteria for mice in all transgenic colonies and murine transplant models included lethargy, impaired motility, splenomegaly, enlarged lymph nodes, decrease in body weight (>20%), development of tumor masses, ruffled fur, hunched back, failure to nest, and loss of appetite. All veterinary

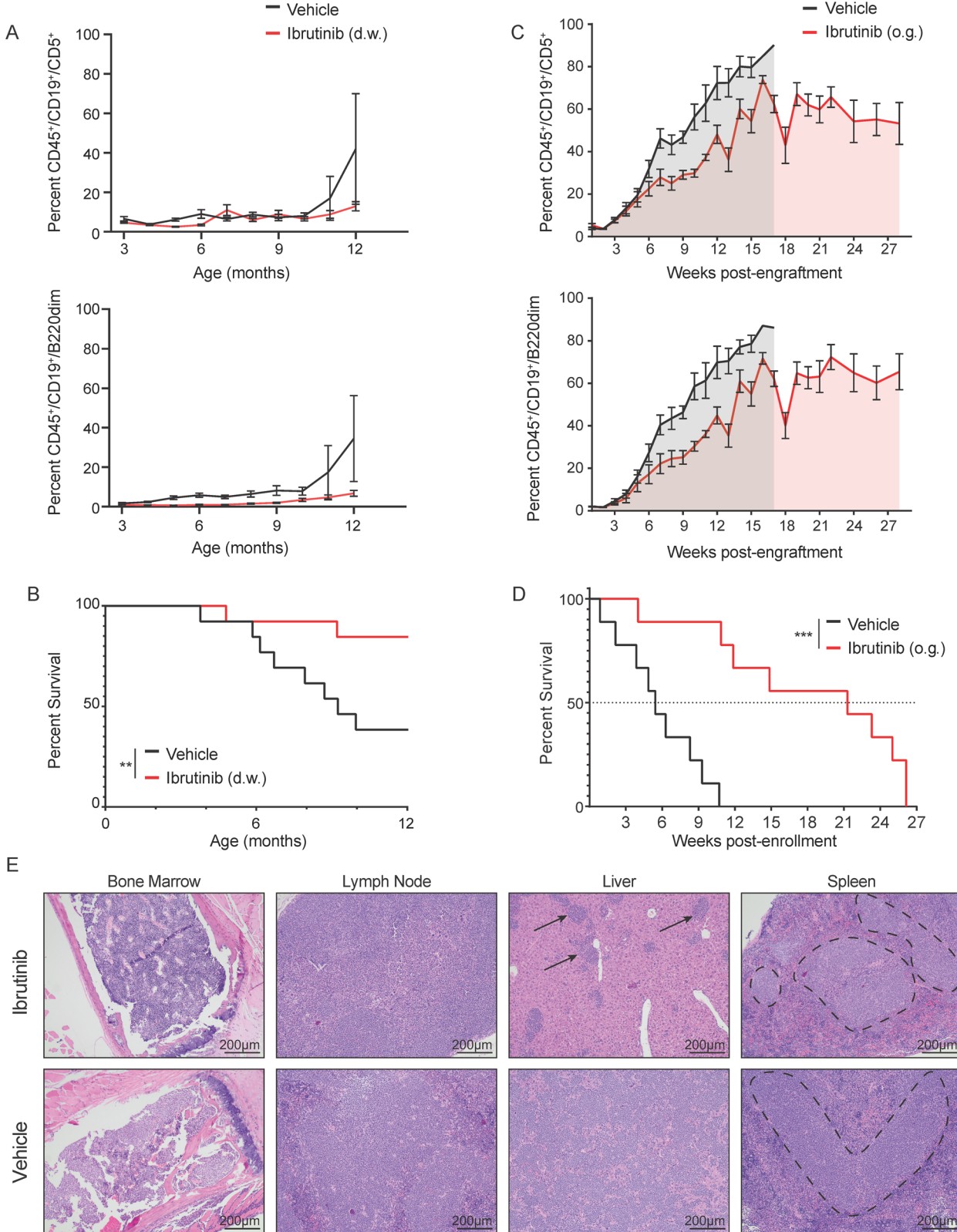

technicians determining removal criteria were blinded to transgenic strain and treatment group.

Transgenic mouse mapping and gene integration services were provided by Taconic Biosciences via a collaboration between Taconic and Cergentis. Sample preparation was carried out at Taconic Biosciences and both TLA and data analysis were carried out at Cergentis as described previously[30], with genotyping assay recommendations subsequently provided by Taconic. The *MTCP1* transgene

integrated at chr8:58,833,688–58,838,745 in an intron of the Galntl6 gene (Founder Line Z20). The region in the genome between the 5′ and 3′ integration site was deleted during the integration event. No structural variations were detected in the transgenic sequence of this sample and >10 copies of the transgene have integrated in this locus. Similarly, the *MTCP1* transgene integrated at chr7:100,715,866–100,720,642 within an intron of the Fam168a gene (Founder line Z36). The genomic region between the 5′ and 3′ integration site was deleted during

**Fig. 6 MTCP1-driven murine leukemia is responsive to BTK inhibition. A** Continuous BTK inhibition via ibrutinib delays progression of a spontaneous leukemia in Eμ-MTCP1 mice most evident by 12 months of age (CD19$^+$/CD5$^+$, $p = 0.018$; CD19$^+$/B220$^{dim}$, $p = 0.012$ determined via Cox proportional hazards model). Eμ-MTCP1 mice ($n = 13$ per group) began continuous dosing of ibrutinib (~30 mg/kg/day, red line) or vehicle (10% cyclodextrin, black line) via drinking water (d.w.) beginning at two months of age. Plot reflects mean ± SE. **B** Mice from (**A**) were followed, revealing continuous BTK inhibition via ibrutinib (red line) prolongs survival compared to vehicle (black line, $p = 0.006$). "**" represents estimation from a Cox proportional hazards model. **C** Adoptive transfer of $1 \times 10^6$ splenic Eμ-MTCP1 CLL-like cells via tail vein to immune competent wildtype host mice results in a progressive leukemic expansion that is delayed upon treatment with ibrutinib. Mice were enrolled to receive either ibrutinib (red line, 25 mg/kg; $n = 9$) or vehicle (black line, 0.5% methylcellulose/1% Tween80; $n = 9$) via daily (Mon-Fri) oral gavage (o.g.) upon reaching >20% CD5$^+$/CD19$^+$ and CD19$^+$/B220$^{dim}$ CD45$^+$ cells in the blood. Ibrutinib administration delayed the rate of leukemic progression measured at six ($p = 0.063$) and 12 weeks ($p < 0.001$) post-engraftment. $P$ values determined via mixed effects model. Plot reflects mean ± SE. **D** Median survival time for mice in (**C**) is significantly extended upon daily administration of oral ibrutinib (red line, median survival = 28.4 weeks), extending well beyond the median survival observed with vehicle treatment (black line, median survival = 11.4 weeks; $p < 0.001$). "***" represents estimation from a Cox proportional hazards model. **E** Post-mortem histopathology analysis of organs and tissues from mice in (**C**) receiving either ibrutinib (top panel) or vehicle (bottom panel) having succumbed to disease. Hematoxylin & eosin staining reveals extensive neoplastic infiltrates effacing the bone marrow and lymph nodes in both ibrutinib and vehicle treated mice, while infiltrates are less severe in the liver and spleen of mice treated with ibrutinib. Neoplastic cells are observed as discrete, basophilic nodules in the livers (arrows) and spleens (circled regions) of the ibrutinib-treated mice, whereas in the vehicle-treated mice, neoplastic cells tended to form diffuse sheets or larger/confluent nodules All organs visualized at 10x, scale bars are 200 μm.

the integration event. No structural variations were detected in the transgenic sequence, and >10 copies of the transgene have integrated in this locus.

**Immunophenotyping.** Due to the high concordance between spleen and lymph nodes, immunophenotyping results are depicted for the spleen. Immunophenotyping of tumor cells in peripheral blood, spleen, and lymph node of Eμ-MTCP1 and Eμ-TCL1 mice by flow cytometry was performed as follows: APC rat anti-mouse CD45 (1/100 dilution; BD Biosciences Cat #559864), FITC rat anti-mouse CD45R/B220 (1/50 dilution; BD Biosciences Cat #553088), BV421 rat anti-mouse CD19 (1/50 dilution; BD Biosciences Cat #562701), PE rat anti-mouse CD5 (1/100 dilution; BD Biosciences Cat #553023). Immunomodulatory assessment of Eμ-MTCP1 peripheral blood by flow cytometry was performed as follows: PE hamster anti-mouse CTLA4 (1/16 dilution; BD Biosciences Cat #553720), PE hamster IgG1 κ isotype control (1/16 dilution; BD Biosciences Cat #553972), BV421 rat anti-mouse CD5 (1/385 dilution; BD Biosciences Cat #562739), FITC rat anti-mouse CD45 (1/50 dilution; BD Biosciences Cat #553080), Alexa Fluor® 647 rat anti-mouse CD19 (1/135 dilution; BD Biosciences Cat #557684), LIVE/DEAD™ fixable near-IR dead cell stain (1/165 dilution; ThermoFisher Scientific Cat #L34976), rat anti-mouse CD16/CD32 (1/100 dilution; Mouse BD Fc Block™; BD Biosciences Cat #553142), BV510 hamster anti-mouse CD3e (1/150 dilution; BD Biosciences Cat #563024), BV650 rat anti-mouse CD11-b (1/200 dilution; BD Biosciences Cat #653402), BB515 rat anti-mouse CD19 (1/200 dilution; BD Biosciences Cat #564509), BUV737 rat anti-mouse CD5 (1/150 dilution; BD Biosciences Cat #612809), APC rat anti-mouse CD93 (1/100 dilution; Biolegend Cat #136510), BUV395 rat anti-mouse CD45R/B220 (1/200 dilution; BD Biosciences Cat #563793), BV786 rat anti-mouse IgM (1/100 dilution; BD Biosciences Cat #564028), PerCP-Cy5.5 rat anti-mouse CD21 (1/200 dilution; Biolegend Cat #1234160), BV711 Rat anti-mouse CD23 (1/200 dilution; BD Biosciences Cat #563987), BV605 Rat anti-mouse IgD (1/100 dilution; BD Biosciences Cat #563003). Immunophenotyping was conducted using Beckman Coulter Gallios 3-laser-10-color (B5-R3-V2) cell analyzer and BD LSRFortessa Cell Analyzer (Cat #649225). Peripheral blood from Eμ-MTCP1 and Eμ-TCL1 transgenic mice was collected monthly via check punch. Cells from the spleens of Eμ-MTCP1 and Eμ-TCL1 having met predefined euthanasia criteria were processed and isolated using methods previously described. Mouse B cells were isolated from whole spleen suspensions using EasySep™ mouse pan B cell isolation kit (STEMCELL Technologies; Cat #19844). All flow cytometry data were analyzed using KALUZA v2.0 software (Becton Dickinson).

Gating strategies followed published data and technical resource publications[59–61] and were adapted to allow exclusion and interrogation of CD19$^+$CD5$^+$ CLL-like populations. Fluorescence-minus-one (FMO) controls were used for each marker and gate position. Cells were gated on viable single mononuclear cells.

**Histopathology.** To define the morphologic characteristics of tumor cells populating Eμ-MTCP1 mice, we performed histopathologic analysis on spleen, thymus, liver, mesenteric lymph node, and bone marrow. Organs were harvested from Eμ-MTCP1 mice meeting predefined euthanasia criteria and representative tissue samples were randomly selected from a pool of mice having evidence of a CLL-like disease. Tissues were fixed in 10% neutral buffered formalin (NBF). Bones were decalcified in formic acid (Surgipath). All tissues were embedded in paraffin and sectioned at 4 μM onto glass slides. Stained sections were assessed by veterinary anatomic pathologists blinded to transgenic strain or treatment group (BH and JC). Staining with hematoxylin and eosin (H&E; Leica) F4/80, B220, and CD3 IHC were

described previously[62]. Photographs were taken using an Olympus SC30 camera with an Olympus BX53 microscope.

**RNA-sequencing.** Cell pellets were captured and washed in PBS on ice prior to resuspension in TRIzol reagent and stored at −80°. Total RNA was isolated from TRIzol suspensions using a chloroform/ethanol extraction method and quantified via Qubit RNA HS Assay kit (Invitrogen). The Clontech SMARTer v4 kit (Takara Bio USA, Inc.) was used for global preamplification. Illumina sequencing libraries were derived from the resultant cDNA using the Illumina Nextera XT DNA Library Prep Kit following manufacturer's instructions. RNA-sequencing libraries were prepared with the Illumina Tru-Seq stranded kit and sequenced on a Hiseq 4000 targeting $40 \times 10^6$ fragments per sample. Transcript-level abundances were estimated using Salmon[63] with the gencode mouse release 23, imported using tximport[64], with normalization and differential expression computed with DESeq2[65]. Data processing was performed according to the CLEAR workflow[66], which identifies reliably quantifiable transcripts in low-input RNA-seq for differentially expressed gene (DEG) transcripts using gene coverage profiles. MiXCR (v3.0.5)[67] was used with default parameters except the RNA-seq alignment was replaced with kaligner2 to identify preprocessed reads containing CDR3 regions from B-cell heavy, kappa, and lambda chains, generating a list of unique CDR3 sequences associated with their relative abundances and specific V(D)J gene usage. MiXCR then generates a list of unique CDR3 sequences associated with their relative abundances and specific V(D)J gene usage. To verify expression of the human MTCP1 and TCL1 transgene (hMTCP1 & hTCL1) in mice, transcript level abundances were estimated using Salmon with a modified gencode mouse reference that contained sequences from human MTCP1 and human TCL1 genes extracted from the grch38 human reference.

**Therapeutic dosing in Eμ-MTCP1 mice.** Eμ-MTCP1 littermate mice were randomized and enrolled to receive continuous ibrutinib (~30 mg/kg/day via drinking water) or vehicle administration beginning at 2 months of age orally via supplemented drinking water. Mice were followed for leukemia onset and overall survival until reaching predefined removal criteria.

Adoptive transfer studies were conducted using $1 \times 10^6$ viable Eμ-MTCP1 splenocytes injected via tail vein to immune competent C57BL/6NTac mice. Engrafted mice were monitored for leukemic expansion via weekly flow cytometry of peripheral blood collected by cheek punch. Upon reaching >20% CD45$^+$/CD19$^+$/CD5$^+$ cells in peripheral blood mice were randomized and enrolled to receive either ibrutinib [25 mg/kg daily oral gavage (o.g.)] or vehicle (0.5% methylcellulose/1% Tween80 o.g.). Continued leukemia progression was monitored by weekly flow cytometry analysis of peripheral blood until reaching predefined removal criteria.

**Statistics.** Unless otherwise noted, analyses were performed by independent statisticians within the OSU Center for Biostatistics according to methods from previously described models. All analyses were performed using SAS/STAT software, version 9.4 (SAS Institute, Inc., Cary, NC). Evaluation of the difference in mean gene expression between cell types collected from the Blueprint DCC gene expression portal generated a two-tailed $p$ value using an unpaired $t$ test with Welch's correction.

For patient data, associations between MTCP1 expression grouped by quartile and demographic, clinical, and molecular features were assessed using Fisher's exact and Kruskal–Wallis tests. MTCP1 expression was correlated with PFS using a Cox stratified proportional hazards model, stratified on study cohort. Further modeling was performed controlling for other important demographic, clinical, or molecular variables. Multiple imputation estimated missing data and combined results for 20 datasets[68]. All $p$ values were two-sided and $p < 0.05$ were considered statistically

significant. No control for multiple comparisons were made. The correlation between *MTCP1* expression and PFS was visualized using Kaplan–Meier plots, grouping patients into quartiles according to *MTCP1* expression.

For mouse survival experiments, survival curve estimates for both overall survival and time to disease onset were calculated using the Kaplan–Meier method and differences in curves were initially assessed using the log-rank test. Next, hazard ratios (HR) and 95% CI were obtained from Cox proportional hazards models to evaluate differences between founder lines/Eμ-TCL1 mice or treatment groups. Mixed effects models were used to assess changes in disease burden over time. Where applicable, data were log-transformed to reduce skewness.

**Illustrations**. Artistic renderings were created and exported under a paid subscription with Biorender.com. Biologic assembly of the proposed 3-dimensional structure of MTCP1 and TCL1 proteins, as determined by x-ray diffraction protein crystallography, was visualized and exported under a University supported subscription using the PYMOL Molecular Graphics System, Version 2.3.5. Unless otherwise noted, data were visualized using GraphPad Prism version 8.3.1 for Windows, GraphPad Software, San Diego, California USA.

**Reporting summary**. Further information on research design is available in the Nature Research Reporting Summary linked to this article.

## Data availability

The RNA-sequencing data generated in this study have been deposited in the GEO database under accession code #GSE176094. The *MTCP1* expression and CLL patient outcomes data, from CALGB studies "10101" and "9712" used in this study, are not publicly available but can be accessed from the authors of these studies (CALGB "9712"- NCT00003248; CALGB "10101" - NCT00098670)[27,28]. All other source data are provided as a Supplementary File. Source data are provided with this paper.

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

## Acknowledgements

We are grateful to the patients who provided blood for the above studies. Research support was provided in part by the National Cancer Institute of the National Institutes of Health (JSW: NIGMS T32 GM068412 and NCATS TL1 TR002735; JC: T32 CA009338). The content is solely the responsibility of the authors and does not necessarily represent the official views of the College of Medicine at The Ohio State University or the National Institutes of Health. This work was also supported in part by the Genetically Engineered Mouse Modeling Core (GEMMC) at the Ohio State University (funded by The OSU Comprehensive Cancer Center Support Grant P30 CA016058) for generation of the Eμ-MTCP1 mouse model. The Ohio State University Comparative Pathology & Mouse Phenotyping Shared Resource (funded by a Cancer Center Support Grant P30 CA016058) supported pathology studies. Transcriptomic library generation and sequencing were supported by The Ohio State University Genomics Shared Resource (supported in part by a Cancer Center Support Grant P30 CA016058). Computational resources were provided by Ohio Supercomputer Center. This work was supported by the National Cancer Institute (R50 CA211524-03 and U54 CA217297 (PY), R35 CA197734 (JCB), R01 CA214046 (RL)). Support from Four Winds Foundation also supported this effort. This study makes use of data generated by the BLUEPRINT Consortium. A full list of the investigators who contributed to the generation of the data is available from www.blueprint-epigenome.eu. Funding for the project was provided by the European Union's Seventh Framework Program (FP7/2007–2013) under grant agreement no 282510 – BLUEPRINT.

## Author contributions

J.S.W. and Z.A.H. designed and conducted experiments, generated data and figures, analyzed data, interpreted results, and wrote the paper. S.S., J.C., K.W., J.N.S., C.B.C., C.T.G., A.P., M.Y., and P.Y. conducted experiments and generated figures. J.C. and B.H. prepared and interpreted histopathology data. L.P.B. and B.R.W. performed animal experiments. Z.A.H., K.W., V.C., J.S.B., and R.L. contributed to the generation of the mouse model. J.M.L. and N.A.H. conducted and interpreted FISH analysis. K.M. and J.A.W. provided and interpreted CLL patient data. A.S.R., A.M.L., and H.G.O. conducted statistical analysis. J.S.B., J.C.B., and R.L. planned the project, acquired funding, supervised the study, interpreted results, and reviewed the paper. These senior authors (J.S.B., J.C.B., R.L.) contributed equally. All authors read and approved the final version of the paper.

## Competing interests

J.S.B. has performed consulting for AbbVie, AstraZeneca, Innate, and KITE Pharma. J.C.B. performed consulting for AstraZeneca, Takada, Novartis, Pharmacyclics, Syndex, and Trillium. J.C.B. chairs the scientific board of Vincera Pharmaceuticals and has significant equity in this company. All other authors declare no competing interests.
