## [Peer Review File · Nature Communications]

Rare t(X;14)(q28;q32) translocation reveals link between MTCP1 and chronic lymphocytic leukemiaREVIEWER COMMENTS

Reviewer #1 (Remarks to the Author):

Summary:

The authors start from a rare t(X;14)translocation in CLL, identifying the translocation partners, then showing an association between higher expression of the translocation partner MTCP1 and CLL outcome, suggesting a role for the gene even outside of the rare translocation. Finally they then go on to generate a transgenic murine CLL model using an overexpression approach, placing the MTCP1 gene under the control of an E μ -enhancer.

This murine model resembles human CLL and an established transgenic murine CLL model (E μ -Tcl-1 transgenics) closely (as far as phenotyped) and shows transplantable tumor cells, with an option to evaluate clinically relevant therapeutics in vivo in a murine model.

The paper is interesting and overall well written, the research is well-designed and thoroughly controlled. The datasets are very complete. Still a few points should be addressed.

Major comments:

In Figures 1 the authors present the relatively higher expression of MTCP1 in CLL cells compared to normal B cells and the quantitative association of MTCP1 expression levels with outcomes in patient cohorts. The discussion should at least provide some educated speculation on what the regulation of MTCP1 might be. Maybe there is some option to interrogate public domain epigenetics databases. Also a discussion of potential upstream signals might be helpful for the readers. Finally, the legend of Figure 1d leaves somewhat unclear whether this is an analysis of MTCP1 translocated CLLs or just CLL in general (which is clear in Fig 1 e)

In the transgenic model the analysis of two founder strains shows relevant difference in the tumor-types observed. The Z20 strain seems to have a relevant percentage of T cell tumors arising. This is not entirely surprising, since the E μ Enhancer is known to be leaky into the T cell compartment in a mouse strain specific manner (the first E μ Bcl2 strains generated in the 90s showed this phenomenon very clearly) – also the MTCP1 is a known T cell oncogene.

This makes the 2 strains difficult to compare. A competing risk analysis, with censoring of mice at time of T cell tumors, and calculation of a murine CLL specific survival, may be able to tease out whether the differences in survival may be due to different kinetics of the non-CLL phenotypes.

The close relationship and potential functional similarity of MTCP1 and Tcl1 – as observed in T cell malignancy and as presented from structural data, together with the close similarities of transgenic mice with both genes is very interesting and leads to a couple of interesting questions (at least interesting to this reviewer).

The apparent functional similarities between MTCP1 and TCL1 suggest potential redundancies. It would be interesting to see whether MTCP1 transgenics have downregulated Tcl1 (and vice versa Tcl1 transgenics with MTCP1 expression). Since the data (at least on mRNA levels) should be included in the RNAseq dataset, this should be an easy question to answer.

Since both MTCP1 and TCL1 seem to functionally converge onto AKT signaling (at least in part) it would be interesting to be able compare the mouse strains regarding the activity along this pathway. At least a pathway centered analysis of the RNA seq should be doable, but also some phosphoproteomics might be feasible.

Minor comments:

The introduction is slightly ambiguous, if not misleading regarding the presentation of the role of Bcl-2. In lines 86ff the might improve the presentation by more clearly stating, that most of the Bcl2

dysregulation is generally attributed to the deletion of mir15/16 in del13q and that a true Bcl2 translocation is a relatively rare event in CLL.

The authors present structural models, but no methodology is clearly stated – the might want to better characterize the source of the dataset.

The transgenic mouse approach makes a relatively good point that the CMC4 gene is merely a passenger in the translocation and might not be essential for the transforming event. While this is circumstantial evidence it might still be helpful for some readers to spell this out.

CLL clones with initial minority have been shown to rapidly grow out to majority clones in transplantation experiments in Tcl1 mice. This should be controlled for especially in treatment studies, since such a phenomenon may generate very different growth kinetics.

Reviewer #2 (Remarks to the Author):

The authors start with the observation of a t(X;14)(q28;q32) translocation in CLL and proceed to define the potential importance of the MTCP1 gene in CLL, including generating a mouse model which develops a CLL-like disease. It is a very nice study and I have only a few comments.

1) In the PFS prognostic analysis of the trial cohorts, does MTCP1 expression stand up in multivariable analysis (I see this in the supplement but missed discussion of it)? In particular, in a bivariate analysis if IGHV status is taken into account? IGHV is a well known predictor of gene expression, does MTCP1 expression associate with IGHV status?

2) In Figure 1a-c, it is hard to see the relevant signals, these should be labeled with arrows. In particular in 1a-b the translocated chromosome is hard to see

3) Better explanation of the CD19 B220 dim subset is required in the text

4) Fig 3A could have littermate comparison spleens

5) Fig 5A – it looks like the IGHV gene usage is different in the MTCP1 vs TCl1 transgenics. Can the authors comment on that at all? in that vein, in 5B, are the ones that segregate together in PC space sharing the same IGHV gene or is there some other similarity they can discern?

6) Table 1 - the DLBCL with the MYC desert translocation partner –did that patient also have CLL?

7) Do the authors have any idea why there is such a difference in their two transgenic strains in terms of time of onset of disease?

8) The authors could consider commenting on the relatively small decrease in peripheral disease burden with ibrutinib, in relation to the more dramatic prolongation in survival (which is similar to what is seen in patients receiving ibrutinib)

Response to Reviewers:

We have received and carefully reviewed the comments to our manuscript entitled “Rare t(X;14)(q28;q32) translocation reveals link between MTCP1 and chronic lymphocytic leukemia.” We appreciate the time the reviewers have taken to review this manuscript and provide feedback, and the manuscript has been strengthened as a result of the comments provided. In this document, we will address these comments in a point-by-point fashion, and we hope the revised manuscript will be acceptable for publication.

Reviewer 1:

The authors start from a rare t(X;14)translocation in CLL, identifying the translocation partners, then showing an association between higher expression of the translocation partner MTCP1 and CLL outcome, suggesting a role for the gene even outside of the rare translocation. Finally, they then go on to generate a transgenic murine CLL model using an overexpression approach, placing the MTCP1 gene under the control of an E μ -enhancer. This murine model resembles human CLL and an established transgenic murine CLL model (E μ -Tcl-1 transgenics) closely (as far as phenotyped) and shows transplantable tumor cells, with an option to evaluate clinically relevant therapeutics in vivo in a murine model. The paper is interesting and overall well written, the research is well-designed and thoroughly controlled. The datasets are very complete. Still a few points should be addressed.

Major comments:

1. In Figure 1 the authors present the relatively higher expression of MTCP1 in CLL cells compared to normal B cells and the quantitative association of MTCP1 expression levels with outcomes in patient cohorts. The discussion should at least provide some educated speculation on what the regulation of MTCP1 might be. Maybe there is some option to interrogate public domain epigenetics databases. Also, a discussion of potential upstream signals might be helpful for the readers. Finally, the legend of Figure 1d leaves somewhat unclear whether this is an analysis of MTCP1 translocated CLLs or just CLL in general (which is clear in Fig 1e).

- *We thank the reviewer for suggesting this opportunity to enhance the discussion of this project. We agree, given the evidence we provide for MTCP1 as an oncogene in B cell leukemia, understanding mechanisms involved in regulating MTCP1 expression is of considerable interest. Indeed, exploring public domain epigenetics databases we identified evidence via transcription factor ChIP experiments in the GM12878 lymphoblastoid cell line demonstrating cis-regulatory elements at the MTCP1 transcription start site with binding of BCL3, ETS1, and IKZF1 (see image #1). However, given the proximity of the transcription start site for the BRCC3 gene and without additional 4C experiments to support our claims – which the authors believe to be beyond the immediate scope of this study – it is impossible to ascertain whether CMC4 and MTCP1 are transcriptionally regulated by these factors. Additionally, we were not able to identify obvious long-range interactions partners with the MTCP1 and CMC4 locus (see image #2). While BCL3, ETS1, and IKZF1 have all been shown to influence leukemia and lymphoma in numerous literature sources, without providing our own transcription factor ChIP experiments in chronic lymphocytic leukemia cells we do not believe these data are appropriate for the presented manuscript.*

(Image #1)

(Image #2)

- However, we agree that the discussion will be improved by providing the readers some educated insight as to how these mechanisms may take place. To address this comment we have expanded the discussion, and included refs #43-46 to support these claims, to now read:

“... Notably, increased *MTCP1* expression in **this CLL cohort** lacked major correlation with pre-treatment characteristics but was associated with a shorter response to chemoimmunotherapy. This relationship may reflect the current understanding of *TCL1* expression in CLL, where inter-patient variability remains the norm yet patients with the lowest *TCL1* expression maintain favorable outcomes following chemoimmunotherapy^{41,42}. **Further correlating *MTCP1* expression with PFS, a bivariate analysis identified a 2-fold increase in *MTCP1* expression as a prognostic indicator of PFS independent of *IgHV* status, a known high-risk factor in CLL. Importantly, however, *IgHV* mutation status as a single variable was not prognostic for reduced PFS in this chemoimmunotherapy cohort – which is atypical in CLL – suggesting further large-scale correlative studies are necessary to provide a comprehensive assessment of the associations between *MTCP1* and other high-risk factors in CLL.**

In a multivariate analysis, elevated *MTCP1* expression lost strong association with reduced PFS while adjusting for additional predictors for reduced PFS in these chemoimmunotherapy trials (Zap-70 methylation, high-risk cytogenetics, sex, WBC). This result implicates *MTCP1* as a factor with considerable influence on the CLL disease course yet may not act as the fundamental driving element, although not entirely surprising considering the abundance of evidence conferring the significant relationship between CLL outcomes and these other high-risk factors. Even so, revealing mechanisms supporting *MTCP1* upregulation in CLL may provide meaningful insight to the overall pathogenesis of this disease. Similar to studies evaluating *TCL1* in CLL⁴³⁻⁴⁵, the apparent inter-patient variability suggests multiple factors likely contribute to dysregulation of *MTCP1* expression; where loss of microRNA-mediated negative regulation or *MTCP1* upregulation via stimulating signals from the tumor microenvironment are primary candidates for further evaluation. Likewise, exploring BCR-induced kinase cascades, with particular attention given toward the role for PKC- β ⁴⁶, may facilitate understanding of the upstream mechanisms upregulating and activating *MTCP1* in CLL. Other mechanisms supporting *MTCP1* upregulation, such as loss of epigenetic control or evasion of X chromosome inactivation (although we observed an even male:female distribution between *MTCP1* expression quartiles) serve as additional unexplored means facilitating *MTCP1* upregulation in CLL. In addition, exploration of this gene in other diseases such as acute myeloid leukemia (AML) might be considered based upon the recent identification of t(X;17)(q28;q21) rearrangement resulting in a *KANSL1-MTCP1* fusion gene in an AML patient⁴⁷.”

- We also apologize for presenting information in Figure 1D that may be unclear to readers. To address this concern we have updated the associated text in the results section, reading:

“We then evaluated *MTCP1* mRNA expression in CLL-B cells **without any known Xq28 rearrangements** and found ~2 fold higher *MTCP1* mRNA transcripts in **these** CLL cells compared to naïve- or memory-B cells (Figure 1D).”

- And updated the associated figure legend, reading:

“(d)*MTCP1* mRNA expression is elevated in CLL cells (**without any known Xq28 rearrangements**) compared to naïve- or memory-B cell subsets (p=0.0012, p=0.0013, respectively). Data obtained from the Blueprint database. CLL cells, n=7; naïve B cells, n=9; memory B cells, n=5. p-value estimated using an unpaired t-test with Welch’s correction.”

2. In the transgenic model the analysis of two founder strains shows relevant difference in the tumor-types observed. The Z20 strain seems to have a relevant percentage of T cell tumors arising. This is not entirely surprising, since the E μ Enhancer is known to be leaky into the T cell compartment in a mouse strain specific manner (the first E μ Bcl2 strains generated in the 90s showed this phenomenon very clearly) – also the *MTCP1* is a known T cell oncogene. This makes the 2 strains difficult to

compare. A competing risk analysis, with censoring of mice at time of T cell tumors, and calculation of a murine CLL specific survival, may be able to tease out whether the differences in survival may be due to different kinetics of the non-CLL phenotypes.

- We thank the reviewer for making this suggestion to improve our manuscript. We agree that the survival comparisons made in our submission were unfair to present alone due to the relevant percentage of T cell tumors in the Z20 founder line. To address this concern we conducted a competing-risk assessment between Z36, Z20 and TCL1 mice with censoring of relevant mice at the time a T cell or myeloid cell abnormality was observed. Presented in Supplemental Figure 3D is a cumulative incidence plot demonstrating the CLL-specific survival estimation of Z36, Z20 and TCL1 mice. We again observed a delayed estimated median survival time in Z20 mice compared to Z36 mice. However, again using only mice that developed a CLL-like disease, when calculating the time from leukemia onset to death we observed similar disease course kinetics between Z20 and Z36 mice (Supplemental Figure 3E).

- Taken together, this evidence leads us to believe that the aggressive CLL-like phenotype present in $E\mu$ -MTCP1 mice remained similar between two unique founder lines and supports the notion that overexpression of MTCP1 in the B cell compartment has significant leukemogenic consequences. To present this analysis we have expanded the results section to read:

“Evaluating only mice that developed a CLL-like phenotype by censoring animals at the time at which a T cell or myeloid cell abnormality was observed, a competing risk assessment estimated the median survival for founder lines Z36 and Z20 to be 12.4 months and 17.6 months, respectively (Supplemental Figure 3D). While the estimated median survival of $E\mu$ -MTCP1 founder Z20 extended beyond that observed in $E\mu$ -TCL1 mice (14.1 months), we observed a significant reduction in time from leukemia onset to death in both Z36 and Z20 $E\mu$ -MTCP1 founder lines (4.10 months and 2.94 months, respectively) when compared to the $E\mu$ -TCL1 model (7.41 months; Supplemental Figure 3E). Taken together, this evidence suggests that while the rate of CLL-like leukemia onset in Z20 lineage mice was significantly delayed in relation to Z36 lineage mice, upon CLL onset the kinetics of this CLL-like disease remained largely similar between the two founder lines.”

- To expand on this idea in our discussion, we have included the following:

“The CLL-like population in $E\mu$ -MTCP1 mice bears a striking resemblance to the disease that develops in $E\mu$ -TCL1 mice; however, the timeline for clinical deterioration from CLL-onset to death was accelerated in both $E\mu$ -MTCP1 founder strains. The observed disparity in median estimated survival between $E\mu$ -MTCP1 founder strains Z20 and Z36 appears to be largely driven by the delayed rate of leukemia onset in Z20 lineage mice. While we confirmed >10 copies of the MTCP1 transgene were inserted in both founder lines, we acknowledge the integration site occurred at two distinct locations on different chromosomes for each respective founder. While mechanisms supporting the observed variance in leukemia onset between Z20 and Z36 founder lines remain unresolved, insertion of the MTCP1 transgene in proximity to additional active enhancer regions or inaccessible in regions of condensed chromatin may be contributing to this distinction. Shared between founder lines, however, the cause

of death for a majority of E μ -MTCP1 mice was attributed to disruption of critical organ function due to systemic malignant infiltration comprised primarily of mixed populations of small-to-intermediate-sized lymphocytes and larger histiocytoid cells. Similar to what is often observed in CLL patients, the spleens of E μ -MTCP1 mice were deformed, consistently presenting with substantial splenomegaly.”

- *The figure legend for Supplemental Figure 3D, 3E, and 3F has been updated to reflect these changes with an amendment to the legend to 3E to address a typing error, reading:*

“(d) Competing-risk assessment to estimate the median survival in E μ -MTCP1 (Z36 and Z20) and E μ -TCL1 mice. Mice were censored (black mark on curve) at the time at which a T cell or myeloid cell abnormality were observed. The median estimated survival time was 12.4 months for E μ -MTCP1 founder Z36 (n=54) and 17.6 months for E μ -MTCP1 founder Z20 (n=36). The Median estimated survival for E μ -TCL1 mice was 14.1 months (n=33). “**” represents p<0.001 and “*” represents p<0.05 using an unpaired t-test with Welch’s correction.**

(e) Evaluation of time between CLL onset (defined in b) and time of death in E μ -MTCP1 (Z36, n=38; Z20, n=10) and E μ -TCL1 (n=32) mice. Reduced time from disease onset to death reflects the delayed onset but rapid disease course in E μ -MTCP1 mice. “**” represents p<0.001 using an unpaired t-test with Welch’s correction.**

(f) Intracellular CTLA-4 expression in CLL-like cells from E μ -MTCP1 mice (founder Z36) resembles that of human CLL. CTLA-4 staining and analysis of peripheral blood cells via flow cytometry reveals elevated intracellular expression in E μ -MTCP1 mice while lacking surface expression (n=4).”

3. The close relationship and potential functional similarity of MTCP1 and Tc11 – as observed in T cell malignancy and as presented from structural data, together with the close similarities of transgenic mice with both genes is very interesting and leads to a couple of interesting questions (at least interesting to this reviewer). The apparent functional similarities between MTCP1 and TCL1 suggest potential redundancies. It would be interesting to see whether MTCP1 transgenics have downregulated Tc11 (and vice versa Tc11 transgenics with MTCP1 expression). Since the data (at least on mRNA levels) should be included in the RNAseq dataset, this should be an easy question to answer.

- *We agree with the reviewer that the similarities between transgenic mouse lines and the apparent functional similarities between MTCP1 and TCL1 lead to several very interesting questions, especially relating to how potential redundancies may downregulate or upregulate the other. To address this question we interrogated the RNA-seq dataset generated from E μ -MTCP1, E μ -TCL1, and wildtype mice, and amended the Salmon alignment protocol to include human MTCP1 and TCL1 gene transcripts in the estimation of transcript abundance (hMTCP1 and hTCL1, respectively). In E μ -MTCP1 mice we observed no change in transcript abundance for the mouse TCL1 gene compared to what was observed in wildtype mice, and in E μ -TCL1 mice we observed no change in transcript abundance for the mouse MTCP1 gene compared to what was observed in wildtype mice. This was indeed a very interesting result and we thank the reviewer for this suggestion. No observable change in expression for MTCP1 and TCL1 in these respective transgenic models compared to wildtype mice suggests the MTCP1-driven leukemia acts independently from TCL1 and the TCL1-driven leukemia acts independently from MTCP1, overall supporting our claim that MTCP1 is a significant leukemogenic factor for CLL. To address these findings, we have updated the results section to read:*

“.... Specifically, **38** genes were identified as overrepresented in E μ -TCL1 mice, highlighted by the presence of various **nucleotide binding factors (Atrip, Ddx60, Trim6)**, **heat shock proteins (Hspa1a/b)**, and other known oncogenic markers (**Adm, Eef2k, Il12a, Ly6i, Mapk8, Map3k20, Pmaip1, Wnt16**). Relative over-expression of **41** genes was noted in E μ -MTCP1 mice, marked by the presence of cell signaling molecules (**Ccr10, Lrp5, Zbtb4**) and other known oncogenic markers (**Cd34, Ptgs1**). **Notably, no change in TCL1 transcript abundance was observed between E μ -MTCP1 and wildtype mice, and no change in MTCP1 transcript abundance was observed between E μ -TCL1 and wildtype mice (Supplemental Figure 5C), suggesting**

the *TCL1*-driven murine leukemia acts independently from *MTCP1* and the *MTCP1*-driven murine leukemia acts independently from *TCL1*.”

- We have also expanded the discussion to include this result, reading:

“...Although the overall transcriptome profile remained largely similar between E μ -MTCP1 and E μ -TCL1 tumor cells, further investigation to define the significance in variation between MTCP1- and TCL1-driven CLL is warranted. Specifically, elucidating the leukemogenic mechanisms which ultimately led to an accelerated disease course from CLL-onset to death in E μ -MTCP1 mice is of considerable interest but is beyond the scope of the present study. Furthermore, the conserved structure between MTCP1 and TCL1A proteins and similarities in the resulting murine and human CLL phenotype suggest shared activation of leukemogenic pathways; **pathways which may be independently activated as no observable change in *TCL1* expression was found in E μ -MTCP1 mice and no observable change *MTCP1* expression was found in E μ -TCL1 mice.**”

- We have also amended all data figures representing the RNA-sequencing data to reflect this new analysis including *hMTCP1* and *hTCL1* genes, and have added panel Supplemental Figure 5C:

F

Supplemental Figure 5

- We have updated the figure legend for Supplemental Figure 5, reading:

“(c) Differential mRNA expression analysis of splenic B cells from Eμ-TCL1 and wildtype mice visualized via volcano plot (n=3 per group). 335 genes enriched in wildtype mice (blue) and 445 genes enriched in Eμ-TCL1 mice (yellow) are shown (Log2FC >2; p-value <0.001). Differential expression of the *MTCP1* gene is highlighted (red).”

- *We have also updated the methods section to reflect this analysis:*

“RNA-Sequencing

Cell pellets were captured and washed in PBS on ice prior to resuspension in TRIzol reagent and stored at -80°. Total RNA was isolated from TRIzol suspensions using a chloroform/ethanol extraction method and quantified via Qubit RNA HS Assay kit (Invitrogen). The Clontech SMARTer v4 kit (Takara Bio USA, Inc.) was used for global preamplification. Illumina sequencing libraries were derived from the resultant cDNA using the Illumina Nextera XT DNA Library Prep Kit following manufacturer’s instructions. RNA-sequencing libraries were prepared with the Illumina Tru-Seq stranded kit and sequenced on a HiSeq 4000 targeting 40x106 fragments per sample. Transcript-level abundances were estimated using **Salmon⁶³ with the gencode mouse release 23, imported using tximport⁶⁴, with normalization** and differential expression computed with DESeq2⁶⁵. Data processing was performed according to the CLEAR workflow⁶⁶, which identifies reliably quantifiable transcripts in low-input RNA-seq for differentially expressed gene (DEG) transcripts using gene coverage profiles. MiXCR (v3.0.5)⁶⁷ was used with default parameters **except the rnaseq alignment was replaced with kaligner2** to identify preprocessed reads containing CDR3 regions from B-cell heavy, kappa, and lambda chains, generating a list of unique CDR3 sequences associated with their relative abundances and specific V(D)J gene usage. **MiXCR then generates a list of unique CDR3 sequences associated with their relative abundances and specific V(D)J gene usage. To verify expression of the human *MTCP1* and *TCL1* transgene (*hMTCP1* & *hTCL1*) in mice, transcript level abundances were estimated using Salmon with a modified gencode mouse reference that contained sequences from human *MTCP1* and human *TCL1* genes extracted from the grch38 human reference.”**

4. **Since both *MTCP1* and *TCL1* seem to functionally converge onto AKT signaling (at least in part) it would be interesting to be able compare the mouse strains regarding the activity along this pathway. At least a pathway centered analysis of the RNA seq should be doable, but also some phosphoproteomics might be feasible.**

- *We thank the reviewer for this suggestion to improve our manuscript, and we agree that interrogating activity along specific pathways will be essential to describing the similarities and differences between the resulting leukemia in Eμ-*MTCP1* and Eμ-*TCL1* mouse models. To address this question, we conducted a pathway centered analysis on genes we found to be significantly enriched in Eμ-*MTCP1* and Eμ-*TCL1* mice compared to wildtype littermates. This analysis revealed that genes shared between the transgenic models primarily converged on G protein-related signaling pathways, likely suggesting that *MTCP1* is indeed (at least in part) acting as an AKT activator. Interestingly, genes uniquely enriched in the Eμ-*MTCP1* model were found to converge on Rho family GTPase signaling, and protein kinase A/cAMP-mediated signaling, providing some evidence that despite the abundance of similarities between models *MTCP1* overexpression is likely acting through additional mechanisms to drive this leukemia. Without comprehensive evidence to support the activation or downregulation of certain pathways, which the authors deem to be beyond the immediate scope of the presented manuscript, we feel it would be inappropriate to comment on this finding any further. To address this question, we have updated the results section to read:*

“...We next sought to determine the overall transcriptional profile of these monoclonal tumor cells. Principal component analysis of the global transcription profiles revealed significant segregation of splenic-B cells **collected** from Eμ-*MTCP1* and Eμ-*TCL1* mice and splenic-B cells from wildtype littermates across PC1 (**Figure 5B**). Further, a significant overlap in gene expression most variable from wildtype splenic-B cells was found when evaluating Eμ-*MTCP1* or Eμ-*TCL1* transgenic strains, suggesting the leukemic B1a cells in Eμ-*MTCP1* mice have achieved a transformed state by both phenotypic and transcriptomic standards (**Figure 5C, D**). **Ingenuity pathway analysis (IPA) of significantly enriched genes (Log2FC > 2, p-value<0.001) shared between Eμ-*MTCP1* and Eμ-*TCL1* transgenic strains when compared to wildtype littermates primarily converged on genes involved in G protein-related signaling pathways (Supplemental Figure 5A), likely a consequence from both *MTCP1* and *TCL1* acting as activators of AKT. In genes uniquely enriched in Eμ-**

MTCP1 mice compared wildtype littermates, IPA analysis converged on FAK and PTEN signaling, Rho family GTPase signaling, and protein kinase A/cAMP-mediated signaling (Figure 5E). Directly comparing Eμ-MTCP1 and Eμ-TCL1 mouse transcriptomes revealed a considerable degree of similarity, where only 79 of 15,318 analyzed genes displayed significant variation from one transgenic model to the other (Figure 5F, Supplemental Figure 5B)...

- *We have also expanded the discussion to read:*

“Although the overall transcriptome profile remained largely similar between Eμ-MTCP1 and Eμ-TCL1 tumor cells, further investigation to define the significance in variation between MTCP1- and TCL1-driven CLL is warranted. Specifically, elucidating the leukemogenic mechanisms which ultimately led to an accelerated disease course from CLL-onset to death in Eμ-MTCP1 mice is of considerable interest but is beyond the scope of the present study. Furthermore, the conserved structure between MTCP1 and TCL1A proteins and similarities in the resulting murine and human CLL phenotype suggest shared activation of **AKT and other** leukemogenic pathways; **pathways which may be independently activated as no observable change in TCL1 expression was found in Eμ-MTCP1 mice and no observable change MTCP1 expression was found in Eμ-TCL1 mice....**”

- *This additional data has been visualized in Figure 5 and a newly created Supplemental Figure 5.*

Supplemental Figure 5

- The associated Figure legends now read:

“Figure 5

(e) Ingenuity pathway analysis (IPA) of significantly enriched genes ($\text{Log}_2\text{FC} > 2$, $p\text{-value} < 0.001$) unique to Eμ-MTCP1 vs wildtype mice or Eμ-TCL1 vs wildtype mice. Rainbow color scale reflects $-\text{Log}(p\text{-value})$.

(f) Differential mRNA expression analysis of splenic B cells from Eμ-MTCP1 and Eμ-TCL1 mice visualized via volcano plot ($n=3$ per group). 92 genes enriched in Eμ-MTCP1 mice (red) or Eμ-TCL1 mice (yellow) are shown ($\text{Log}_2\text{FC} > 2$; $p\text{-value} < 0.001$).”

“Supplemental Figure 5

(a) Ingenuity pathway analysis (IPA) of significantly enriched genes ($\text{Log}_2\text{FC} > 2$, $p\text{-value} < 0.001$) shared between Eμ-MTCP1 and Eμ-TCL1 transgenic strains when compared to wildtype littermates. Rainbow color scale reflects $-\text{Log}(p\text{-value})$.”

Minor Comments:

1. The introduction is slightly ambiguous, if not misleading, regarding the presentation of the role of Bcl-2. In lines 86ff they might improve the presentation by more clearly stating, that most of the Bcl2 dysregulation is generally attributed to the deletion of mir15/16 in del13q and that a true Bcl2 translocation is a relatively rare event in CLL.

- *We agree with the reviewer and apologize for presenting misleading information to the reader. To address this comment, we have revised the corresponding paragraph in the introduction to now read:*

“...Despite their infrequency, molecular profiling of these rare rearrangements have revealed broad importance of previously un-recognized coding or non-coding genes critical to the pathogenesis of CLL. Practical application of this strategy **facilitated understanding the role of the anti-apoptotic protein BCL2 in CLL. While abundantly present in follicular lymphoma and diffuse large B cell lymphoma (DLBCL), the t(14;18)(q32;q21) translocation – involving the IGH locus and the BCL2 gene – is a rare event in CLL (1-2% of cases)^{9,10}. Yet even in absence of a t(14;18)(q32;q21) rearrangement, it was found that BCL2 mRNA was over-expressed in virtually all CLL patients compared to normal B cells¹¹. Work to determine mechanisms driving this abnormality later revealed microRNAs *miR-15/16* as leading posttranscriptional regulators of BCL2 and loss of *miR-15/16* as a result of 13q14 deletions substantially associate with elevated BCL2 expression in CLL¹²⁻¹⁴. The pathogenic importance of this discovery subsequently led to a greater understanding of CLL disease biology and mechanisms resulting in the dramatic clinical activity demonstrated by the BH3-mimetic venetoclax¹⁵; ultimately contributing to the overall improvement in the therapeutic management of CLL.”**

- *We have also adjusted a similar statement in the discussion to now read:*

“...A broader and more general finding relevant to other types of cancer is the successful application of a strategy pursuing the functional consequence of genes involved in rare chromosomal abnormalities. **Exploring the rare t(14;18)(q32;q21) translocation in CLL, such approaches facilitated understanding the significance of the anti-apoptotic protein BCL2 and miR-15/16** at the minimal deleted region of del(13q14)¹²⁻¹⁴. These **microRNAs** contribute to the pathogenesis of CLL, and BCL2 represents an exceptionally valuable therapeutic target¹⁵.”

2. The authors present structural models, but no methodology is clearly stated – they might want to better characterize the source of the dataset.

- *We agree and thank the reviewer for bringing this to our attention. Accordingly, we have updated relevant text in the results section to read:*

“Previous studies, **using x-ray diffraction to estimate the** crystal structure of both p13 MTCP1 and p14 TCL1A proteins¹⁴⁻¹⁶, describe a high degree of overlap between both amino acid sequence and 3-dimensional protein conformation – highlighted by an eight-strand beta barrel tertiary structure remarkably unique to this family of proteins...”

- *Similarly, we have updated relevant text in the methods section under the heading ‘Screening suspected CLL cases for Xq28 rearrangements’ to read:*

“**The proposed 3-dimensional crystal structure** for MTCP1 and TCL1A protein sequences were obtained from the RCSB Protein Data Bank (IDs: 1A1X, 1JSG, **respectively**)^{15,16}. **Crystal structures were determined via x-ray diffraction and presented at 2.00 Å and 2.50 Å resolution, respectively.**”

- *We have also updated relevant text in the methods section under the heading ‘Illustrations’ to read:*

“Artistic renderings were created and exported under a paid subscription with Biorender.com. **Biologic assembly of the proposed 3-dimensional structure of MTCP1 and TCL1 proteins, as determined by x-ray diffraction protein crystallography**, was visualized and exported under a University supported subscription using the PYMOL Molecular Graphics System, Version 2.3.5.”

3. The transgenic mouse approach makes a relatively good point that the CMC4 gene is merely a passenger in the translocation and might not be essential for the transforming event. While this is circumstantial evidence it might still be helpful for some readers to spell this out.

- *We agree with this point and thank the reviewer for suggesting we provide a more robust discussion to the readers. We have expanded our discussion to include this topic, adding ref #56 to support our claims, reading:*

“...The transgenic strategy of the E μ -MTCP1 mouse presented herein also supports the notion that the *CMC4* gene is merely a passenger in the t(X;14)(q28;q32) rearrangement and may not provide an essential role in the leukemogenic transforming event. Regardless, *CMC4* expression was indeed elevated in CLL cells when compared to normal B cell subsets. When translated to the short p8 MTCP1 isoform this protein is localized to the mitochondria⁵⁶, likely suggesting an indirect role in supporting leukemogenesis via metabolic pathways. Thus, complete investigation of the relationship between p8 and p13 MTCP1 may further define the collective oncogenic impact of Xq28 rearrangements.”

4. CLL clones with initial minority have been shown to rapidly grow out to majority clones in transplantation experiments in Tc11 mice. This should be controlled for especially in treatment studies, since such a phenomenon may generate very different growth kinetics.

- *We thank the reviewer for this comment. We agree that minority clones rapidly growing out to majority clones in transplant experiments may generate different growth kinetics in treatment studies and should indeed be taken into consideration. For this reason, we designed our study to control for this. To summarize the experiment represented in figure 6C and 6D, whole spleen cell suspension from one donor E μ -MTCP1 mouse was engrafted into recipient mice. Recipient mice were monitored for evidence of successful engraftment via flow cytometry of the blood on a weekly basis, and mice were enrolled into treatment groups when the percentage of CD19+/CD5+ cells in the CD45+ compartment reached a threshold of 20%. Mice were randomly assigned to groups until n=9 had been reached for each group. Specifically, at the time of each weekly flow cytometry evaluation for enrollment, mice were assigned to treatment group on a rolling and alternating basis to ensure even distribution of mice with rapidly growing and slow growing tumors were present in each group. Thus, an even number of mice with high percentage of CD19+/CD5+ cells (>30%) and low percentage of CD19+/CD5+ cells (~25-20%) at the time of the weekly monitoring were assigned to each group.*
- *To further demonstrate that this concern was controlled for in our study, we have conducted a new analysis plotting the survival of the first three mice assigned to each group vs the final six mice assigned to each group. This analysis is shown below, demonstrating that regardless of the time of enrollment the survival time was significantly extended in mice treated with ibrutinib although the extended survival was reduced in mice enrolled at a later point suggesting a different growth kinetic of those later tumors as the reviewer pointed out hence the importance of controlling these type of studies. We have discussed this in the results section of the revised manuscript:*

“...We further evaluated the use of the E μ -MTCP1 adoptive transfer model for pre-clinical evaluation of CLL drug candidates. Successfully engrafted mice **with comparable disease load (percent CD19+/CD5+ B cells in the blood) were randomly assigned to receive ibrutinib or vehicle by daily oral gavage **at any given enrollment time to control for different growth kinetics of the engrafted tumor cells**. Similarly, ibrutinib administration led to a reduction in the rate of disease development between six and 12 weeks post-enrollment and prolonged survival compared to those receiving vehicle (**Figure 6C, D**)...”**

Reviewer 2:

The authors start with the observation of a t(X;14)(q28;q32) translocation in CLL and proceed to define the potential importance of the MTCP1 gene in CLL, including generating a mouse model which develops a CLL-like disease. It is a very nice study and I have only a few comments.

1. In the PFS prognostic analysis of the trial cohorts, does MTCP1 expression stand up in multivariable analysis (I see this in the supplement but missed discussion of it)? In particular, in a bivariate analysis if IGHV status is taken into account? IGHV is a well-known predictor of gene expression, does MTCP1 expression associate with IGHV status?

- We thank the reviewer for suggesting this opportunity to improve the assessment presented in our manuscript. Presented in Table 1, we identified an even distribution of patients with mutated or unmutated IgHV among all patients when stratified into quartiles by MTCP1 expression, likely suggesting the apparent association between elevated MTCP1 expression and reduced PFS is independent of IgHV status. However, to address this comment, we conducted a bivariate analysis with IgHV mutation status and MTCP1 expression as a continuous variable and found that a 2-fold increase in MTCP1 expression was indeed prognostic for PFS independent of IgHV status. This data has been added to the manuscript as Supplemental Table 2, reading:

Table S2: Stratified proportional hazards models* for progression-free survival

Variable	Stratified Models for Each Variable	Stratified Model including MTCP1 Expression and IgHV ^b
	p (HR, 95% CI)	p (HR, 95% CI)
MTCP1 Expression, 2-fold increase	0.03 (2.09, 1.08-4.05)	0.03 (2.08, 1.07-4.05)
IgHV, Unmutated vs Mutated	0.67 (1.12, 0.66-1.91)	0.87 (1.05, 0.61-1.80)

*Stratified on study cohort.

*Each predictor is adjusted for all others in the model.

HR = hazard ratio, CI = confidence interval

- We have updated the results section to reference this analysis, reading:

“...To interrogate the clinical significance of *MTCP1* mRNA expression in CLL, we conducted a retrospective analysis on *MTCP1* expression in CLL patients from two independent chemoimmunotherapy trial study cohorts for which microarray data have been previously reported (CALGB ‘9712’ and ‘10101’)^{27,28}. Here, we correlated CLL risk factors with *MTCP1* expression using baseline characteristics obtained at time of treatment initiation (**Table 2**). When stratified into quartiles by *MTCP1* expression we observed a similar distribution **between sexes and high-risk CLL cofactors including age, performance status, cytogenetic evaluation, *IgHV* mutation status, and Zap-70 methylation**; with the exception of elevated blood lymphocyte counts (WBC) **and advanced Rai stage at diagnosis** skewing towards patients with higher *MTCP1* expression (Q2-4). **As a single continuous variable, a 2-fold increase in *MTCP1* expression was also found to associate with shorter progression-free survival (PFS) when adjusting for study cohort (p=0.03; Supplemental Table 1)**. To illustrate this association, we plotted PFS by *MTCP1* expression quartile for all patients and for each study cohort; visualizing that patients with the lowest expression (Q1) tended to have longer PFS (**Figure 1E; Supplemental Figure 1D, E**). **Further, as *IgHV* mutation status is a strong predictor of outcome in CLL¹, a bivariate analysis including continuous *MTCP1* expression and *IgHV* status found a 2-fold increase in *MTCP1* expression was prognostic for PFS independent of *IgHV* status (p=0.03; Supplemental Table 2).”**

- *We have also expanded our discussion of this, reading:*

“... Notably, increased *MTCP1* expression in **this CLL cohort** lacked major correlation with pre-treatment characteristics but was associated with a shorter response to chemoimmunotherapy. This relationship may reflect the current understanding of *TCL1* expression in CLL, where inter-patient variability remains the norm yet patients with the lowest *TCL1* expression maintain favorable outcomes following chemoimmunotherapy^{41,42}. **Further correlating *MTCP1* expression with PFS, a bivariate analysis identified a 2-fold increase in *MTCP1* expression as a prognostic indicator of PFS independent of *IgHV* status, a known high-risk factor in CLL. Importantly, however, *IgHV* mutation status as a single variable was not prognostic for reduced PFS in this chemoimmunotherapy cohort – which is atypical in CLL – suggesting further large scale correlative studies are necessary to provide a comprehensive assessment of the associations between *MTCP1* and other high-risk factors in CLL.**

In a multivariate analysis, elevated *MTCP1* expression lost strong association with reduced PFS while adjusting for additional predictors for reduced PFS in these chemoimmunotherapy trials (Zap-70 methylation, high-risk cytogenetics, sex, WBC). This result implicates *MTCP1* as a factor with considerable influence on the CLL disease course yet may not act as the fundamental driving element, although not entirely surprising considering the abundance of evidence conferring the significant relationship between CLL outcomes and these other high-risk factors. Even so, revealing mechanisms supporting *MTCP1* upregulation in CLL may provide meaningful insight to the overall pathogenesis of this disease. Similar to studies evaluating *TCL1* in CLL⁴³⁻⁴⁵, the apparent inter-patient variability suggests multiple factors likely contribute to dysregulation of *MTCP1* expression; where loss of microRNA-mediated negative regulation or *MTCP1* upregulation via stimulating signals from the tumor microenvironment are primary candidates for further evaluation. Likewise, exploring BCR-induced kinase cascades, with particular attention given toward the role for PKC- β ⁴⁶, may facilitate understanding of the upstream mechanisms upregulating and activating *MTCP1* in CLL. Other mechanisms supporting *MTCP1* upregulation, such as loss of epigenetic control or evasion of X chromosome inactivation (although we observed an even male:female distribution between *MTCP1* expression quartiles) serve as additional unexplored means facilitating *MTCP1* upregulation in CLL. In addition, exploration of this gene in other diseases such as acute myeloid leukemia (AML) might be considered based upon the recent identification of t(X;17)(q28;q21) rearrangement resulting in a *KANSL1-MTCP1* fusion gene in an AML patient⁴⁷.”

2. In Figure 1a-c, it is hard to see the relevant signals, these should be labeled with arrows. In particular in 1a-b the translocated chromosome is hard to see.

- We thank the reviewer for bringing this to our attention and apologize for the lack of clarity. Accordingly, we have enlarged the images in Figures 1A-C and added arrows to indicate relevant signals.

Figure 1

3. Better explanation of the CD19 B220 dim subset is required in the text.

- We agree and thank the reviewer for this suggestion to improve the manuscript. To provide a more robust explanation of the CD19⁺/B220^{dim} B cell subset we have revised this section of the results to now read:

“...CLL is characteristically defined as an accumulation of CD45⁺ B lymphocytes with phenotypic expression of cell-surface markers CD5, CD19, and CD23, with dim – or intermediate – expression of CD45R (B220)¹. To evaluate the potential of the E μ -MTCP1 model to develop a leukemia resembling a CLL-like phenotype, littermate mice from Z36 and Z20 founder lines were followed monthly via flow cytometry analysis of the blood. Gating on CD45⁺ cells and probing for CD19⁺/CD5⁺ and CD19⁺/B220^{dim} cell populations, a progressive expansion of circulating CLL-like B cells were detected as early as 5 months of age in both E μ -MTCP1 founder lines (Figure 2A)...”

4. Fig 3A could have littermate comparison spleens.

- We agree with this comment and apologize for omitting this comparison from the original submission. A comparison spleen has been added to Figure 3A, and the associated figure legend has been updated to now read:

“(a) Representative gross images of severely enlarged spleens from Z36 and Z20 founder E μ -MTCP1 mice, an observation consistently observed in subsequent progeny comprising the E μ -MTCP1 colonies. **A representative gross image of a healthy spleen from an age-matched wildtype mouse is shown for comparison. Red arrows indicate location of the spleen in the dissected mouse.**”

Figure 3

5. Fig 5A – it looks like the IGHV gene usage is different in the MTCP1 vs TCL1 transgenics. Can the authors comment on that at all? In that vein, in 5B, are the ones that segregate together in PC space sharing the same IGHV gene or is there some other similarity they can discern?

- We agree that the IgHV usage was found to be different in Eμ-MTCP1 and Eμ-TCL1 mice and is worth commenting on. Accordingly, we have updated the results section to now read:

“... Prominent usage of distinct heavy chain gene loci was observed from splenic B cells isolated from Eμ-MTCP1 mice, a stark contrast from the high degree of clonal variability exhibited in wildtype mice (Figure 5A, Supplemental Table 3). The major clone in 2/3 Eμ-MTCP1 and 3/3 Eμ-TCL1 mice predominantly used V_H1 and V_H12 family genes, while 3/3 wildtype mice predominantly used V_H5 family genes. One analyzed Eμ-MTCP1 mouse used V_H5 family genes in it's dominant clones. The V_H12-3 gene was shared as that predominantly used in the major clone of one Eμ-MTCP1 and Eμ-TCL1 mouse, for which these mice also shared dominant use of V_K and V_L genes. All other Eμ-MTCP1 and Eμ-TCL1 mice displayed different V_H usage in the dominant clone. In general, similar patterns suggesting distinctly clonal leukemic populations were observed by kappa and lambda light chain usage (Supplemental Figure 4A).”

- However, we do not believe the differential IgHV gene usage is driving the major changes we observed between Eμ-MTCP1 and Eμ-TCL1 mice. Upon further inspection requested by the reviewer, we found that the one Eμ-MTCP1 mouse and the two Eμ-TCL1 mice segregating together in PC space did not share the same IgHV, IgKV, or IgLV gene usage in the dominant clone of each respective mouse. Further, there were no obvious similarities to discern from inspecting the RNA-seq dataset for these mice. Regardless, we do think this point raised by the reviewer is important in that the way we have presented this data may be misleading the reader and not effectively communicating the significance of the result. We believe the major significance of this result to be that the spleens in both Eμ-MTCP1 and Eμ-TCL1 mice are primarily comprised by an expansion of clonally related cells, and that this observation significantly varies from what is normally found in wildtype mice. To better illustrate this point, we have updated figure 5A and Supplemental Figure 4A.

Figure 5

Supplemental Figure 4

- We have also updated the associated figure legends to now read:

“Figure 5:

(a) Comparison of IGHV gene usage among E μ -MTCP1 (founder Z36), E μ -TCL1, and wildtype mice (n=3 per group) demonstrated clonal tumor populations in E μ -MTCP1 and E μ -TCL1 mice. To examine differences in variable region (V) gene usage, BCR sequences were binned by specific heavy- (H) and light-chain V genes, normalized by the total number of reads corresponding to that chain. **The percent tumor volume comprised by the dominant clone of each mouse shown in black. All minority clones are grouped and their percent contribution is visualized via grey-checked pattern. Gene names of the top 10 V genes in order of abundance are presented in Supplementary Table S3.**”

“Supplemental Figure 4:

(a) Comparison of IGKV and IGLV gene usage between E μ -MTCP1, E μ -TCL1, and wt mice (n=3 per group) demonstrates a clonal tumor population **in the spleen** of E μ -MTCP1 and E μ -TCL1 mice. BCR reads were binned by heavy chain V gene names, with each V gene normalized by the total number of heavy-chain reads. **The percent tumor volume comprised by the dominant IgK and IgL genes of each mouse are shown in black. Percent usage of all other IgK and IgL genes are grouped and visualized via grey-checked pattern.** Gene names of the top 10 V genes in order of abundance are presented in Supplementary Table S3.

6. Table 1 - the DLBCL with the MYC desert translocation partner –did that patient also have CLL?

- We thank the reviewer for bringing this case to our attention. This patient was seen initially with suspected CLL/lymphoid malignancy which was later characterized as DLBCL. Upon further review of the cytogenetic analysis available for this patient we identified a $t(X;8)(q28;q24.2)$ rearrangement, which is most frequently present in a distinct subset of unusual CLL cases. The inclusion of an Xq28 rearrangement in this patient may be of interest to readers and we have thereby expanded our discussion to include this observation, adding references #53 and #54 to support these claims, reading:

“Microdeletions in the 12q32 site including the HOXC cluster and other adjacent genes results in significant disruption of normal cell function⁴³⁻⁴⁵, and amplification of the 8q22 site including EDD1 and GRHL2 genes,

negative regulators of apoptosis, has been described in breast, pancreas, and lung cancer cells as a mechanism for evasion of death receptor-activated therapies⁴⁶. The 8q24.2 region is a gene desert containing *MYC* enhancer elements and variations at this site are known to influence CLL⁴⁷. **Interestingly, the t(X;8)(q28;q24.2) case reported here was identified in a patient screened initially on the basis of suspected CLL which was later found to be DLBCL. Cytogenetic evaluation of this patient, a 59 y/o male, also identified a t(3;9)(q27;q32) rearrangement involving the *BCL6* gene and a t(14;19)(q32.3;q13.2) rearrangement involving the *BCL3* gene. This t(14;19) rearrangement, joining the *BCL3* locus with *IGH* elements, occurs in DLBCL and other chronic lymphoproliferative disorders but is most frequently observed in CLL. However, t(14;19) CLL cases are often atypical with distinctive clinicopathologic and genetic features including younger age, aggressive clinical course, and association with trisomy 12, which was also present in this patient^{53,54}. Translocation of the *MTCP1* locus under *MYC* enhancer elements in this unusual case supports our finding that, while rare, genomic rearrangement events resulting in up-regulation of the *MTCP1* gene may contribute to the transformative potential of the leukemic B cell and influence the overall trajectory of the resulting tumor burden.**

7. Do the authors have any idea why there is such a difference in their two transgenic strains in terms of time of onset of disease?

- *As mentioned in our response to comment #2 from reviewer 1, we acknowledge that Z36 and Z20 Eμ-MTCP1 founder strains displayed significantly variable rates of leukemia onset. Presently, providing a complete evaluation of mechanisms driving this disparity are beyond the scope of this study. However, we do provide speculation that the difference in disease onset between these two lines may be a result of the MTCP1 transgene inserted in proximity to additional enhancer regions in the Z36 founder line, or the MTCP1 transgene in the Z20 founder line may be partially inaccessible due to its' insertion into regions of condensed chromatin. Regardless, we did observe that while the rate of leukemia onset differed between the two presented founder lines, we observed similar disease course kinetics between both founder lines after the CLL-like leukemia had been established. The authors consider this result a particular strength of our study, reporting that insertion of the MTCP1 transgene to two unique locations in the mouse genome were both capable of driving a CLL-like disease with a similar overall phenotype. To reflect this evaluation, we have expanded our discussion to read:*

“The CLL-like population in Eμ-MTCP1 mice bears a striking resemblance to the disease that develops in Eμ-TCL1 mice; however, the timeline for clinical deterioration from CLL-onset to death was accelerated in both unique Eμ-MTCP1 founder strains. Thus, the observed disparity in median estimated survival between Eμ-MTCP1 founder strains Z20 and Z36 appears to be largely driven by the delayed rate of leukemia onset in Z20 lineage mice. While we confirmed >10 copies of the *MTCP1* transgene were inserted in both founder lines, we acknowledge the integration site occurred at two distinct locations on different chromosomes for each respective founder. While mechanisms supporting the observed variance in leukemia onset between Z20 and Z36 founder lines remain unexplored, insertion of the *MTCP1* transgene in proximity to additional active enhancer regions or inaccessible in regions of condensed chromatin may be contributing to this disparity. Shared between founder lines, however, the cause of death for a majority of Eμ-MTCP1 mice was attributed to disruption of critical organ function due to systemic malignant infiltration comprised primarily of mixed populations of small-to-intermediate-sized lymphocytes and larger histiocytoid cells. Similar to what is often observed in CLL patients, the spleens of Eμ-MTCP1 mice were deformed, consistently presenting with substantial splenomegaly.”

8. The authors could consider commenting on the relatively small decrease in peripheral disease burden with ibrutinib, in relation to the more dramatic prolongation in survival (which is similar to what is seen in patients receiving ibrutinib)

- *We thank the reviewer for suggesting this opportunity to improve the manuscript. To address this comment we have expanded the discussion, and added ref #55 to support our claim, reading:*

“Having demonstrated that the E μ -MTCP1 mouse model generates a CD19+/CD5+ B cell malignancy with resemblance to human CLL, we proposed that this model may be ideally suited for pre-clinical evaluation of novel therapeutic agents for consideration in CLL and related diseases. To support this proposal, we demonstrated that continuous ibrutinib administration in pre-leukemic mice delayed disease onset and prolonged survival. In an adoptive transfer model, daily ibrutinib dosing delayed the rapid advancement of engrafted E μ -MTCP1 splenocytes and resulted in a dramatic prolongation in survival. **The moderate reduction in circulating lymphocytes upon treatment with ibrutinib in this adoptive transfer model is consistent with the understanding that inhibition of BCR signaling via ibrutinib drives cell mobilization from nodal sites and often results in prolonged lymphocytosis⁵⁵. The dramatic prolongation in survival seen here is likely a result of less severe occupation of proliferative lymphoid compartments and defacement of normal organ architecture in engrafted mice. Overall, the observed response to inhibition of BCR signaling encourages further use of this model in development of novel derivatives with varying specificity for BTK and other components of the BCR pathway.**”

REVIEWER COMMENTS

Reviewer #1 (Remarks to the Author):

The authors have addressed all comments satisfactorily.

Reviewer #3 (Remarks to the Author):

The present manuscript by Walker and Lapalombella and their colleagues starts out from the cloning of a novel chromosomal translocation in a case of chronic lymphocytic leukemia (CLL). Compared to normal B cells, the gene which is aberrantly expressed as the result of the translocation, MTCP1, was found to be expressed twice as high in a panel of CLL cases. The historical comparison to BCL2, which was also identified through a very rare translocation in CLL, although BCL2 expression levels are generally elevated in CLL and the importance of BCL2 expression in CLL pathophysiology has been well established, led the authors to investigate the consequences of increased MTCP1 expression in an animal model. In two transgenic lines, CLL-like disease developed besides various other malignancies with a penetrance of 30 to 70 percent, respectively. Clonality was established, and phenotypic analysis of CLL-like lymphoproliferations developing in those mice suggests relatedness to CLL developing in other CLL mouse models, with some minor differences which may reflect the activity of MTCP1. The tumor cells were sensitive to ibrutinib treatment. Overall, the authors propose that elevated MTCP1 expression in CLL contributes to pathogenesis, and that the line which gives rise to a higher fraction of CLL cases may provide a useful preclinical model to test novel drugs for potential therapeutic use.

This reviewer substituted a referee of the primary submission who was unable to see the revision. Although I believe the jury is still out on whether aberrant expression of the MTCP1 gene will turn out to have a role in human CLL pathophysiology or not, the results of the present study are hypothesis generating and provide a clear rationale to follow up such research. The largely positive reviews of the primary submission had brought up a few contentious issues and asked for clarification of several points, which the authors in my view have excellently addressed by providing new data and biostatistical analyses as well as thoroughly discussing the reasons as to why they were unable to address some of the points. Clearly, the study which overall is excellently presented (with perhaps one exception – see below) is of interest for the field. I nevertheless suggest a couple of modifications to enhance clarity:

- 1) I understood the figure panels Fig. 5A and Supplemental Fig. 4A only after reading the responses to the reviewers. The use of the description Mouse I to III using Greek numbering is confusing especially since one gets the impression in the supplemental figure that Mouse I to III show both kappa and lambda rearrangements. Of course, the numbers do not refer to the same mouse but this is not clear. I suggest omitting any numbering here and describe what the circles are in the figure legend.
- 2) The sentence on page 7 that “neoplastic cells exhibited variable degrees of F4/80 and B220 expression...” F4/80, really? How is this possible? Or is the “neoplastic cells” here the wrong subject of the sentence? Please clarify.
- 3) I think Fig. 6E would improve by pointing out with arrows where to look at to see the differences, and perhaps by actually describing in the figure legend what one sees in the organs with and without treatment. It is clear to me but perhaps not to every reader.

Response to Reviewers:

We have received and carefully reviewed the comments after primary revision to our manuscript entitled “Rare t(X;14)(q28;q32) translocation reveals link between MTCP1 and chronic lymphocytic leukemia.” We appreciate the time the reviewers have taken to review this manuscript and provide feedback, especially grateful for newly added reviewer #3, and the manuscript has been strengthened as a result of the comments provided. In this document, we will address these comments in a point-by-point fashion, and we hope the final draft of this manuscript will be acceptable for publication.

Reviewer 1:

The authors have addressed all comments satisfactorily.

- We thank this reviewer for carefully considering our revisions to this manuscript.

Reviewer 3:

The present manuscript by Walker and Lapalombella and their colleagues starts out from the cloning of a novel chromosomal translocation in a case of chronic lymphocytic leukemia (CLL). Compared to normal B cells, the gene which is aberrantly expressed as the result of the translocation, MTCP1, was found to be expressed twice as high in a panel of CLL cases. The historical comparison to BCL2, which was also identified through a very rare translocation in CLL, although BCL2 expression levels are generally elevated in CLL and the importance of BCL2 expression in CLL pathophysiology has been well established, led the authors to investigate the consequences of increased MTCP1 expression in an animal model. In two transgenic lines, CLL-like disease developed besides various other malignancies with a penetrance of 30 to 70 percent, respectively. Clonality was established, and phenotypic analysis of CLL-like lymphoproliferations developing in those mice suggests relatedness to CLL developing in other CLL mouse models, with some minor differences which may reflect the activity of MTCP1. The tumor cells were sensitive to ibrutinib treatment. Overall, the authors propose that elevated MTCP1 expression in CLL contributes to pathogenesis, and that the line which gives rise to a higher fraction of CLL cases may provide a useful preclinical model to test novel drugs for potential therapeutic use.

This reviewer substituted a referee of the primary submission who was unable to see the revision. Although I believe the jury is still out on whether aberrant expression of the MTCP1 gene will turn out to have a role in human CLL pathophysiology or not, the results of the present study are hypothesis generating and provide a clear rationale to follow up such research. The largely positive reviews of the primary submission had brought up a few contentious issues and asked for clarification of several points, which the authors in my view have excellently addressed by providing new data and biostatistical analyses as well as thoroughly discussing the reasons as to why they were unable to address some of the points. Clearly, the study which overall is excellently presented (with perhaps one exception – see below) is of interest for the field. I nevertheless suggest a couple of modifications to enhance clarity:

Comments:

- 1) I understood the figure panels Fig. 5A and Supplemental Fig. 4A only after reading the responses to the reviewers. The use of the description Mouse I to III using Greek numbering is confusing especially since one gets the impression in the supplemental figure that Mouse I to III show both kappa and lambda rearrangements. Of course, the numbers do not refer to the same mouse but this is not clear. I suggest omitting any numbering here and describe what the circles are in the figure legend.
- We thank the reviewer for making this suggestion to improve our manuscript. This point further addresses our representation of data in figure 5A and supplemental figure 4A that were previously raised by reviewer #2 in the initial review of this manuscript. As evident from this comment, this data remained unclear to the

reviewer and is likely to distract future readers. Data in figure 5A and supplemental figure 4A demonstrate clonality in the B cell expansion of E μ -MTCP1 and E μ -TCL1 mice, contrasting the polyclonal B cell populations regularly found in wildtype mice. We have updated text in the results section, shown below, to clarify this point associated with figure 5A and supplemental table 3. We further believe that the data in supplemental figure 4A is providing confusion to the readers while not sufficiently contributing to the overall story we present. As such, we have decided to leave figure 5A as is, leaving labels as 'mouse I-III' using Greek numbering to designate the different analyzed animals, and have removed panel A from supplemental figure 4 showing kappa and lambda usage and the associated text in the results section.

"To understand if the aggressive murine leukemia in E μ -MTCP1 transgenic mice were composed of heterogeneous, polyclonal B cells or derived from a single precursor founding a homogeneous tumor population as is the case in human CLL, we evaluated the B cell receptor (BCR) repertoire by examining IGH transcripts via RNA-sequencing methods previously described by our group³⁶. Prominent usage of distinct heavy chain gene loci representing a clonal B cell expansion was observed from splenic B cells isolated from 3/3 E μ -MTCP1 mice and similarly in 3/3 E μ -TCL1 mice, a stark contrast from the high degree of clonal variability exhibited in wildtype mice (Figure 5A, Supplemental Table 3). The major clone in 2/3 E μ -MTCP1 and 3/3 E μ -TCL1 mice predominantly used VH1 and VH12 family genes, while 3/3 wildtype mice predominantly used VH5 family genes. One analyzed E μ -MTCP1 mouse used VH5 family genes in its dominant clones. The VH12-3 gene was shared as that predominantly used in the major clone of one E μ -MTCP1 and E μ -TCL1 mouse. ~~for which these mice also shared dominant use of VK and VL genes. All other E μ -MTCP1 and E μ -TCL1 mice displayed different VH usage in the dominant clone. In general, similar patterns suggesting distinctly clonal leukemic populations were observed by kappa and lambda light chain usage (Supplemental Figure 4A).~~ The most abundant tumorigenic clones isolated from E μ -MTCP1 spleens exhibited a low IGHV mutational burden (Supplemental Figure 4AB), well below the threshold for classification³⁷ as "mutated" (Supplemental Figure 4BC). The low mutational burden in this region is consistent with the aggressive, IGHV-unmutated, subtype of human CLL."

- As suggested, we have updated the figure legend for figure 5A to better describe the data represented in this panel.

"(a) Comparison of IGHV gene usage among E μ -MTCP1 (founder Z36), E μ -TCL1, and wildtype mice (n=3 per group) demonstrated clonal tumor populations in E μ -MTCP1 and E μ -TCL1 mice. ~~To examine differences in variable region (V) gene usage, BCR sequences were binned by specific heavy (H) and light chain V genes, normalized by the total number of reads corresponding to that chain. Visualized via pie chart, the percent tumor volume comprised by the dominant clone of each mouse shown in black. All minority clones are grouped and their percent contribution is visualized via grey-checked pattern. Gene names of the top 10 IGH V genes in order of abundance are presented in Supplementary Table S3.~~

- Supplemental figure 4 now displays as shown below, with the associated figure legend.

Supplemental Figure 4

“Supplemental Figure 4. Low IGHV mutation burden observed in Eμ-MTCP1 and Eμ-TCL1 B lymphocytes

(a) Comparison of IGKV and IGLV gene usage between Eμ-MTCP1, Eμ-TCL1, and wt mice (n=3 per group) demonstrates a clonal tumor population in the spleen of Eμ-MTCP1 and Eμ-TCL1 mice. BCR reads were binned by heavy chain V gene names, with each V gene normalized by the total number of heavy-chain reads. The percent tumor volume comprised by the dominant IgK and IgL genes of each mouse are shown in black. Percent usage of all other IgK and IgL genes are grouped and visualized via grey-checked pattern. Gene names of the top 10 V genes in order of abundance are presented in Supplementary Table S3.

(a) Mutation status of the 20 most abundant IGHV genes in Eμ-MTCP1, Eμ-TCL1, and non-transgenic mice (data shown from one mouse is representative of n=3 per group). The vast majority of IGHV reads in the dominant clones from each mouse revealed a low mutational burden. IGHV gene sequences determined from RNA-sequencing were compared against germline controls to assess mutation frequency. The X-axis represents the analyzed clone from most to least abundant. Y-axis represents mutation frequency per clone.

(b) IGHV gene sequences determined from RNA-sequencing were compared against germline controls to assess mutation frequency in Eμ-MTCP1 and Eμ-TCL1 spleen cells. A low mutation rate in the IGHV region was identified in Eμ-MTCP1 and Eμ-TCL1 cells, well below the threshold (>2%) for classification of “mutated IGHV.”

2) The sentence on page 7 that “neoplastic cells exhibited variable degrees of F4/80 and B220 expression...” F4/80, really? How is this possible? Or is the “neoplastic cells” here the wrong subject of the sentence? Please clarify.

- We thank the reviewer for this suggestion to clarify our manuscript. The reviewer is correct, the subject of this sentence was incorrect and we apologize for this mistake. We have updated this section of the results to now read:

“Eμ-MTCP1 mice meeting ERC due to CLL-like disease invariably presented with splenomegaly accompanied by abdominal lymphadenopathy (Figure 3A). Histopathology evaluation revealed variable neoplastic infiltration of lymphoid tissues including robust splenic and lymphatic involvement with very modest presence in the marrow (Figure 3B). **Tumor cross sections Neoplastic cells** exhibited variable degrees of F4/80 and B220- **expression expressing** infiltrates as detected by immunohistochemistry, with scant numbers of CD3+, mature, well-differentiated lymphocytes, which were interpreted as tumor-associated lymphocytes³⁵. Healthy mouse tissues were infiltrated and effaced by neoplastic populations resembling both small-to-intermediate-sized lymphocytes and larger histiocytoid round cells, similar to those previously reported for the Eμ-TCL1 mouse model³¹.”

- 3) I think Fig. 6E would improve by pointing out with arrows where to look at to see the differences, and perhaps by actually describing in the figure legend what one sees in the organs with and without treatment. It is clear to me but perhaps not to every reader.

- We agree with this comment and have accordingly updated the figure and associated figure legend.

“(e) Post-mortem histopathology analysis of organs and tissues from mice in (c) receiving either ibrutinib (top panel) or vehicle (bottom panel) having succumbed to disease. Hematoxylin & eosin staining reveals **extensive neoplastic infiltrates effacing the bone marrow and lymph nodes in both ibrutinib and vehicle treated mice, while infiltrates are less severe in the liver and spleen of mice treated with ibrutinib. Neoplastic cells are observed as discrete, basophilic nodules in the livers (arrows) and spleens (circled regions) of the ibrutinib-treated mice, whereas in the vehicle-treated mice, neoplastic cells tended to form diffuse sheets or larger/confluent nodules. less-severe organ involvement, specifically in the liver and spleen, in ibrutinib-treated mice compared to vehicle-treated mice.** All organs visualized at 10x, scale bars are 200μm.”